# From Coastal to Montane Forest Ecosystems, Using Drones for Multi-Species Research in the Tropics

**Dede Aulia Rahman \*, Andre Bonardo Yonathan Sitorus and Aryo Adhi Condro**

Department of Forest Resources Conservation and Ecotourism, Faculty of Forestry & Environment,
IPB University, Kampus IPB Dramaga, Bogor 16680, Indonesia;
andre_bonardo@apps.ipb.ac.id (A.B.Y.S.); condroadhi@apps.ipb.ac.id (A.A.C.)
\* Correspondence: dedeaulia@apps.ipb.ac.id

**Abstract:** Biodiversity monitoring is crucial in tackling defaunation in the Anthropocene, particularly in tropical ecosystems. However, field surveys are often limited by habitat complexity, logistical constraints, financing and detectability. Hence, leveraging drones technology for species monitoring is required to overcome the caveats of conventional surveys. We investigated prospective methods for wildlife monitoring using drones in four ecosystems. We surveyed waterbird populations in Pulau Rambut, a community of ungulates in Baluran and endemic non-human primates in Gunung Halimun-Salak, Indonesia in 2021 using a DJI Matrice 300 RTK and DJI Mavic 2 Enterprise Dual with additional thermal sensors. We then, consecutively, implemented two survey methods at three sites to compare the efficacy of drones against traditional ground survey methods for each species. The results show that drone surveys provide advantages over ground surveys, including precise size estimation, less disturbance and broader area coverage. Moreover, heat signatures helped to detect species which were not easily spotted in the radiometric imagery, while the detailed radiometric imagery allowed for species identification. Our research also demonstrates that machine learning approaches show a relatively high performance in species detection. Our approaches prove promising for wildlife surveys using drones in different ecosystems in tropical forests.

**Keywords:** animal behavior; disturbance; forest ecosystem; machine learning; primates; savanna; UAV; waterbird; wildlife conservation

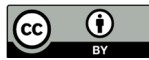

## 1. Introduction

The tropical rainforest is one of the most complex ecosystems. It is also a vital ecosystem, which acts as a biodiversity fortress and provides crucial ecosystem services such as raw materials (e.g., timber and non-timber forest products), soil protection, carbon sequestration and watershed protection [1–3]. Despite representing the largest reservoir of biodiversity with a rich diversity of flora (>200 plant species/hectare) [4,5] and fauna (>50% of the world's animal species) [6] and its various functionalities, most tropical rainforests, including those situated in Indonesia, are lost or degraded through human intervention over time. Biological diversity in Indonesia is the second greatest globally, and Indonesia has the third-largest area of tropical rainforest [7], with a span of 94.1 million ha of forest in 2019 [8]. Nineteen types of natural ecosystems occur here, from lowland rainforest to alpine, and from marine coastal to deep-sea ecosystems [9]. Although Indonesia comprises only 1.3% of the Earth's surface, it houses about 15% of its species richness [7,10].

Two key information challenges inhibit progress on achieving Aichi Targets 11 (Protected Areas) and 12 (Preventing Extinctions). Both challenges involve significant disparities in availability and quality of tropical biodiversity data [11]. Reliable data on wildlife populations are critical for making informed management decisions and, in particular, there is a lack of primary in situ data on populations in tropical protected areas [12], thus

preventing the rigorous evaluation of population responses to threats and drawing conclusions that are counterproductive to targeted conservation efforts [13,14]. Conclusions based on combined secondary data and expert opinion are often wrong, hence more objective evaluation of primary data and strong indicators of change in biodiversity over time [15] is required. Population monitoring programs for tropical wildlife species are routinely and periodically needed to address incomplete knowledge [16,17]. To adequately conserve wildlife species, managers need to set boundaries of acceptable change for species and their habitats, including setting priority areas to preserve the species through analysis of robust data from regular monitoring activities [18].

Wildlife monitoring in tropical rainforest areas presents additional challenges. Some species are primarily elusive, of drab coloration, secretive by habitat, present in small numbers, and generally difficult to observe in the wild due to densely vegetated, rugged and remote habitat conditions. Monitoring strategies based on indirect encounters have been developed by practitioners. These have proven effective in overcoming problems in monitoring some wildlife species [19], though can also be challenging [20–22]. Direct monitoring of wildlife populations using ground-based transect surveys is still widely used [23], however requires intensive fieldwork and is highly dependent on surveyor competence in estimating animal-observer distances and identifying species, which can be difficult in tropical forests with dense vegetation and low light [24]. Clearing and following a planned path can make surveying problematic over rough terrain, can be hard work, time-consuming, and subsequently detrimental to data collection [24]. Moreover, low encounter rates and small sample sizes, typical in endangered species monitoring with transects sampling leads to problems in data analysis [18,25]. Monitoring methods are an essential part of wildlife ecology research for assessing and forecasting changes and species management practices reveal the potential for co-application of proven traditional techniques and modern technologies for better environmental control and management processes.

In recent decades, the use of unmanned aerial vehicles (UAVs, a.k.a drones) has doubled over time; the lack of detection using some traditional techniques can often be overcome through these new techniques, especially for cryptic species living in remote areas [26]. Imagery from aerial surveys has been useful for a variety of wildlife surveys for example, minimizing human disturbance in monitoring endangered raptors [27], detection, behavioral responses assessment, and estimating kangaroo density [28–30], ungulate enumeration with thermal surveys [31,32], Nile crocodile population surveys [33], and so forth. Meanwhile, in tropical areas with thick canopy, monitoring wildlife using drones is still rare because visual detection with standard aerial imagery often does not work in this typical habitat. The use of visible spectrum (RGB) cameras on drones has been shown to work suboptimally in detecting subjects with weak contrast against the background [34], and active in the lower and middle canopies. Recently, the use of thermal-infrared (TIR) cameras that rely on subject detection based on endothermic heat signatures is increasingly being investigated for surveying wildlife species that are difficult to detect using visible spectrum drone imagery for example, [29,35,36]. However, some of the challenges of applying thermal infrared imagery in wildlife monitoring are the low-resolution image of TIR cameras that make it difficult to identify species, regulations limiting drone operations, and high dependence on weather conditions [31].

Potential biases in data collection and the completeness of the research outcomes will be primarily determined by the selection of wildlife monitoring methods used; this is an essential aspect of successful project planning [18]. Furthermore, the selection of the most appropriate method will be determined according to the advantages and disadvantages of each method with regard to constraints on a particular survey, and it is crucial to ensure the most beneficial technique is used to monitor wildlife populations [17]. Here, we report on using a dual visible-thermal camera approach to conduct drone surveys of wildlife species within three protected areas in Indonesia with diverse ecosystem types. We paired drone surveys with traditional ground surveys to assess drone efficacy as a wildlife sur-

vey tool within various ecosystems. The focus of the study on focal animals in each location is shown to overcome the difficulties in monitoring activities for various objectives. We paired drone surveys with traditional ground surveys to compare: (1) nest counts derived from each method, and additionally observed nesting material and breeding of waterbirds to strengthen conservation efforts this species in coastal and mangrove forest in Pulau Rambut Wildlife Sanctuary (PRWS); (2) assessment of species diversity and detection techniques from thermal imagery of two large-sized mammals that inhabit the savanna of Baluran National Park (BNP); and (3) the behavioral response of three primate species in Gunung Halimun Salak National Park (GHSNP) to the drone at various altitudes. We demonstrate the potential of the dual-camera approach while making recommendations to improve such a system for future monitoring of wildlife and conservation improvement in tropical areas and globally.

## 2. Materials and Methods

### 2.1. Study Areas

Our primary motivation was to evaluate the effectiveness of using drones for focal species inhabiting various ecosystem types in tropical forest areas (see in Appendix A). Surveys were conducted in three ecosystems representing the diversity of ecosystems in tropical rain forests in Indonesia, as follows (Figure 1).

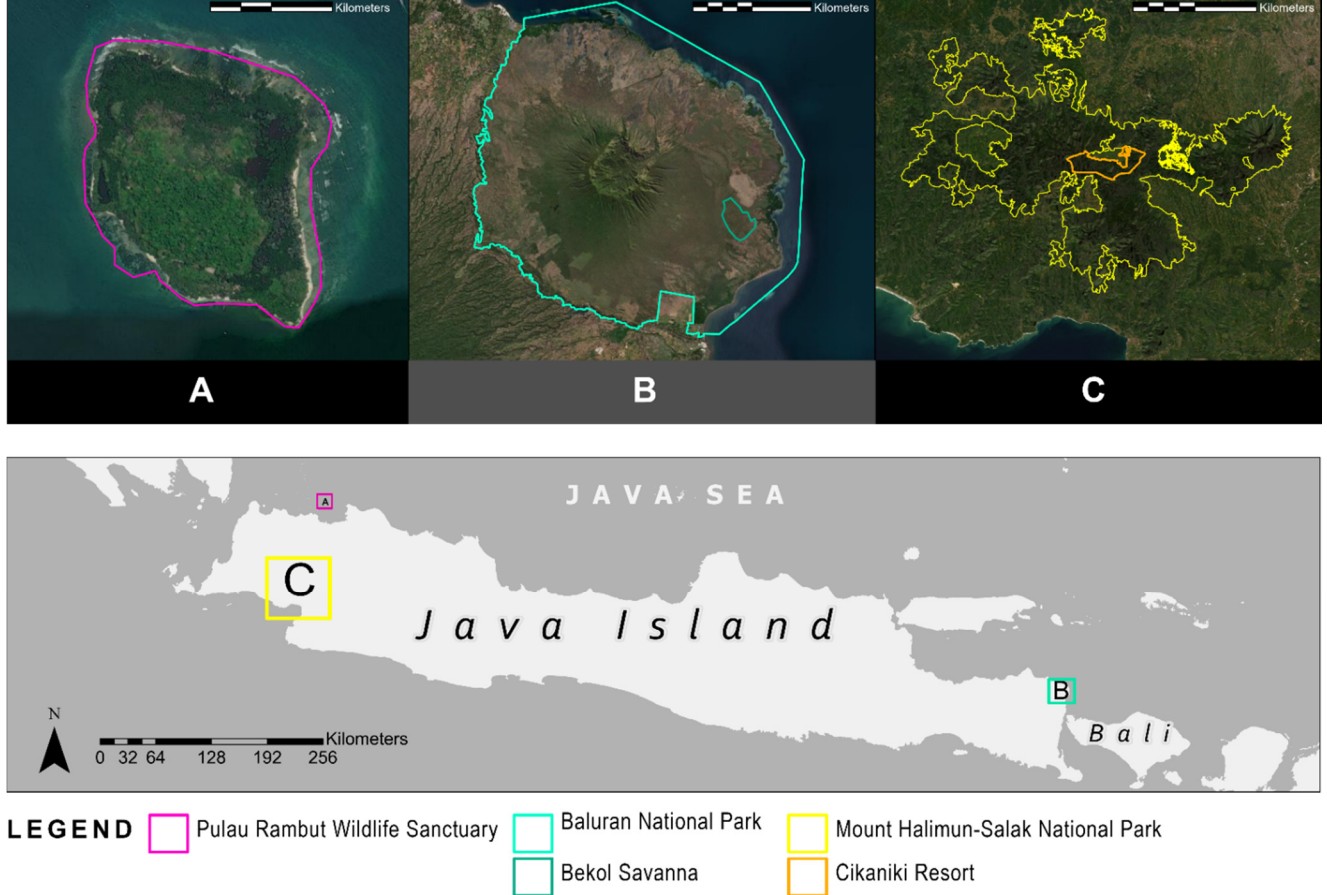

**Figure 1.** Three protected areas in Indonesia where drone surveys with a dual visible-thermal camera system and associated ground surveys were conducted for multi-species research inhabiting: (**A**) coastal and mangrove forest in PRWS, (**B**) savanna in BNP, and (**C**) montane forest in GHSNP. The figure below depicts an inset map of the three ecosystems of our study areas.

### 2.1.1. Pulau Rambut Wildlife Sanctuary (PRWS)

We surveyed PRWS from 24 July to 22 August 2021. PRWS is located in the Jakarta Bay area (106.5°41′30″ E–5.5°58′30″ S) and comprises 45 ha of three types of forest ecosystems in PRWS, including a diversity of wetland habitats ranging from the coastal forest, lowland forest, and mangrove forest. The area is classified as climate type C based on the Schmidt dan Ferguson Classification (Köppen 1936). It has a marked wet season (generally from November to April) and a single dry season (generally from May to October). PRWS is one of seven Ramsar Sites in Indonesia that have an important role in protecting the wetlands ecosystem. Moreover, PRWS is a breeding habitat for about 20,000 individuals from 22 waterbird species and is the breeding site for the milky stork *Mycteria cinerea* in Java Island. Waterbird species found in Pulau Rambut are oriental darter *Anhinga melanogaster*, grey heron *Ardea cinerea*, purple heron *A. purpurea*, great egret *Egretta alba*, little egret *E. garzetta*, plumed egret *E. intermedia*, reef egret *E. sacra*, cattle egret *Bubulcus ibis*, black-crowned night heron *Nyticorax nyticorax*, milky stork *Mycteria cinerea*, pygmy cormorant *Phalacrocorax pygmaeus*, little black cormorant *P. sulcirostris*, pied cormorant *P. melanoleucus*, glossy ibis *plegadis falcinellus*, and black-headed ibis *Threskiornis melanocephalus* [37–39]. All of these species can be grouped into settled and nonresident waterbirds. In addition to bird species, other species of fauna are also found such as *Pteropus vampyrus*, *Varanus salvator*, *Boiga dendrophila*, and *Python sp* [37].

### 2.1.2. Baluran National Park (BNP)

We surveyed BNP from 28 May to 14 July 2021. BNP is located in the East Java Province, Indonesia (74°9′52.3″ S, 114°23′15.4″ E) and covers an area of 25,000 ha of a mixture of tropical deciduous forest with a savanna ecosystem. The park is surrounded by villages, the sea, and teak plantations. The area is classified as climate type AW based on the Köppen Climate Classification [40]. Here the rainy season is very short and normally occurs from December to February, therefore, the lack of water is the most important limiting factor for vegetation in the area. The largest portion of the foot and slopes of Gunung Baluran is covered by semi-open deciduous forest. The forest continuously extends into the lowland area, which is dominated by savanna interspersed with small forests along the galleries, on top of hills, and in humid areas. Initially, almost half of the park was covered by grassland, though this has been altered by an invasive species, *Acacia nilotica*. Species that occur in the study area include *Bos javanicus*, *Bos bubalis*, *Rusa timorensis*, *Muntiacus muntjak*, *Sus scrofa*, *Panthera pardus melas*, *Cuon alpinus*, *Trachypitecus auratus*, *Macaca fascicularis*, *Hystrix javanica*, *Prionailurus bengalensis*, *Prionailurus hermaphroditus*, and *Pavo cristatus* [41].

### 2.1.3. Gunung Halimun Salak National Park (GHSNP)

We surveyed GHSNP from 14 January to 17 March 2021. GHSNP is located in West Java Province, Indonesia (06°44′21″ S 106°31′53″ E and 06°44′47″ S 106°32′1″ E) and comprises 113,357 ha of lowland to montane vegetation. All surveys were conducted in the Cikaniki Resort Research Facility (CRRF) in GHSNP, which is located in the center of the Mount Halimun area at the foot of Mount Kendeng. The area is characterized by a landscape mosaic, dominated by the colline to montane forest ecosystem and classified as 'Colline Primary Forest Zone', interspersed with several settlement areas (enclaves). The vegetation community is dominated by *Homalantus populneus*, *Nauclea lanceolata*, and *Macaranga sp*. Four dominant trees were recorded along the Cikaniki-Citalahab loop trail during the fieldwork; *Lithocarpus javensis*, *Altingia excelsa*, *Litsea sp.*, and *Elaeocarpus sp*. The climate in CRRF is categorized as type A. The mean temperature ranges between 21–25 °C with relative humidity between 72%–89%. Mean annual rainfall and annual temperature range between 3200–6000 mm and 16–30 °C, respectively. The topography at Cikani Resort varies, with altitudes ranging from 500 to 1700 m above sea level with a slope of >15°. CRRF is one of the important areas in Indonesia for primate conservation, and there

are four endemic primates in Java that inhabit this area, *Hylobathes moloch*, *Trachypithecus auratus*, *Presbytis comata*, and *Nycticebus javanicus* [42].

## 2.2. Mission Planning Considerations

We collected all described drone data using a DJI Matrice 300 RTK with Thermal Camera DJI H20T (Table A1 Appendix B) and a DJI Mavic 2 Enterprise Dual (Table A2 Appendix B) operated from an Apple 7.9″ iPad (128 GB, Wi-Fi+4G LTE) using a DJI Pilot app. Both drones have a maximum flight time of 30 min, a maximum speed >70 km h$^{-1}$, a pilot-controlled range of >5 km, and are equipped with a 20 MP camera with high definition 4 K/60 fps video capacity. We programmed all flight plans using Mission Planner (DJI Pilot version 1.9.0 by DJI Innovations Technology Co. LTD, Shenzhen, China) (Figure A1 Appendix C). All images were assembled and ortho-rectified using Agisoft Photoscan Pro ver. 1.2.5.2594, which is now Agisoft Metashape (https://www.agisoft.com/ accessed on 18 September 2021), prior to importing them into QGIS ver. 2.8.6 (QGis Development Team, Beaverton, USA) for analysis.

Before implementing our drone-based multi-species surveys, we tested drones for the effects of disturbance (e.g., escape, attacking, or other evasive maneuvers) that might affect survey results for different species across different types of ecosystems. We conducted a series of fifteen test flights in each study. The DJI Matrice 300 RTK was used to survey coastal and mangrove forests and savanna ecosystems and the DJI Mavic 2 Enterprise Dual in the montane forest to assess the impacts of the drone on disturbance for each species inhabiting each location. We set the minimum flying height without disturbance for each group of species as the height of the last flight before the heights when they fled. Drones were flown for 25 min at each location, starting at 120 m and lowered 10 m every 2 min to assess the level of disturbance (Table A3 Appendix C). We limited the drone's flying height to 120 m following the restrictions on the flying height of unmanned aircraft as regulated in the Regulation of the Minister of Transportation of the Republic of Indonesia No. 37/2020 concerning the Operation of Unmanned Aircraft in Airspace Served by the State of Indonesia.

Furthermore, to optimize detectability in the resulting images, we tested photographic parameters and ambient lighting on various flights performed. Increasing the flying altitude means expanding the coverage area of the survey, reducing the required flight duration, but causing an increase in battery power consumption. However, an increase in flight altitude causes a decrease in the resolution of the resulting photo; meanwhile, the number of photos to be processed is also reduced. We flew three test flight sessions at each location to establish flight and photographic parameters that optimize detection. These three sessions were conducted on the same day and consecutively with 25 min intervals for each session. Fifteen flights in each session were carried out at different altitudes, that is, 10, 20, 30, 40 m and up to 120 m. The resulting maps of the study site were imported into the GIS, and three experienced independent observers carefully detected the presence of each wildlife species on the maps. We limited observers to 10 min per map, and they were blind to the corresponding flight parameters. A fourth observer performed a posteriori recount on all maps without a time limit to estimate the frequency of false detections to verify observer reliability. All analyses considered the time of day of the flight, flight altitude and observer identity as factors influencing the detection of each wildlife species.

## 2.3. Survey Comparison Study

Generally, we compared the effectiveness of drone surveys to traditional wildlife survey protocols, namely, ground-based transect surveys (henceforth referred to as ground surveys), for considerable research with a focus on focal species in each study site. We implemented each of the two survey types successively, following the protocols below, on the same day, starting with a drone survey. We conducted ground surveys in the same area as covered by the corresponding drone flight.

### 2.3.1. Drone Surveys

Following the results of our optimal flight evaluations in each study site (Table A3 Appendix C), we flew drone surveys at an optimum altitude to detect each focal species and at a speed of 5 m s$^{-1}$ with 90° camera orientation, autonomously following a pre-programmed flight plan from take-off to landing. For each site, we repeated the same flight plan three times in the same day (if the logistics allowed), namely, one between 06.00 h and 09.00 h, one between 11.00 h and 14.00 h, and one between 16.00 h and 19.00 h. We programmed the drone to take photos at regular intervals that ensured a minimum 60% overlap between two consecutive images to optimize photo collation and avoid shadows on maps [43]. We made maps from the field survey as described above and visually searched maps to identify species (using head or body shape visible in photographs), quantify the number of nests and observed nesting material and breeding of waterbirds in PRWS, detection of species diversity and identification techniques from thermal photographs of two large-sized mammals that inhabit the savanna of BNP. For the primate community in GHSNP, we specifically assessed the behavioral responses of three primate species to drones at various altitudes.

### 2.3.2. Ground Surveys

We counted waterbird nests in the PWRS, recorded the diversity of wildlife species in the savanna of BNP after drone flights, and searched for all target objects using binoculars. We crossed the entire drone track on foot. Due to logistical issues, we could not always replicate the diurnal survey protocol three times (once per drone count). For each detected nest/animal object, we identified the species and recorded a GPS point of its location. Three observers at the same location walked the transect and used binoculars to monitor the behavior of the various wildlife found at each location.

We explored the montane forest for in GHSNP and searched for individuals/groups of our three focal endemic primate species. Data recording for ground-based surveys and drones was carried out at the beginning of the encounter until the primate individual/group was no longer detected. Data were collected in three time periods with 9 h of observations per day. The three monitoring periods were period I (06:00–09:00), period II (11:00–14:00), and period III (16:00–19:00). All observers monitored 14, 12, and 14 transect surveys in the three sampling periods for Javan gibbon, Javan langur, and Javan surili populations. Three observers were included in each survey to reduce the possibility of observer bias.

### 2.4. Data Analysis

### 2.4.1. Nest Counts and Observed Nesting Material and Breeding of Waterbirds in PRWS

We created a shapefile containing a grid using ArcGIS to overlay the image to help with nest calculations. Grid cells were 10 m × 10 m, with a total of 4500 grid cells that divide the study area. One independent observer who was not involved in the field surveys analyzed all images to avoid bias and performed a manual and visual identification of the nests 'by eye' in the orthomosaic. When calculating nests in an image, we examined the radiometric (RGB) imagery for the grid cells created. Then we tagged each species-specific shapefile by making a dot at each presumed nest in the RGB imagery. We also utilized TIR imagery to confirm suspected nests in the RGB imagery and to detect and confirm the nest's location, which was not directly visible in the RGB imagery. Next, we categorize the dot at each presumed nest in the RGB imagery and divide it into empty nests, filled nests, and non-nests. Due to the difficulty of distinguishing empty and filled nests through ground surveys, we only recorded nest findings as depicted by a green dot in Figure 2. In the post-processing phase on RGB imagery, everything is seen in the nest (nesting materials, eggs, chicks and birds) were assigned a number and counted by an observer using the recorded videos on a personal computer using the free software GIMP for image processing (release 2.10; https://www.gimp.org accessed on 27 September 2021).

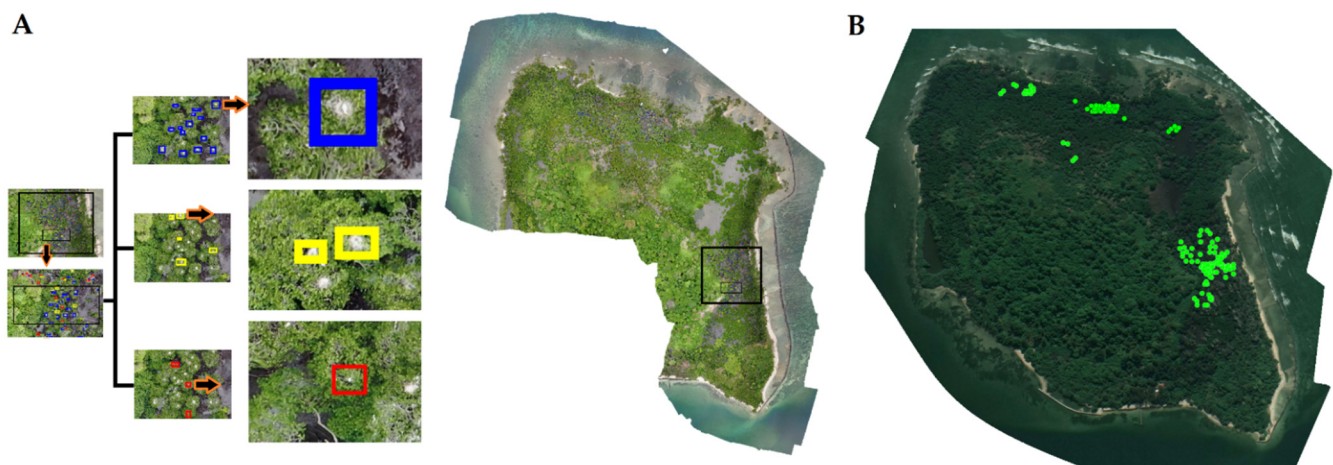

**Figure 2.** (**A**) Orthomosaic image of high-resolution of the nests by drone and categorized into filled nests (blue box), empty nests (yellow box), and non-nests (red box), and (**B**) presence of nests from the ground survey (green dot).

2.4.2. Rapid Assessment of Species Diversity, Species Detection and Model Evaluation in BNP

We visually identified each species from ground surveys and recorded all the species derived from RGB imagery. We chose four general indexes to measure the composition of wildlife species found in the savanna ecosystem and assess the feasibility of using drones for this purpose. Three measures of species diversity were used in this calculation, including: (1) the species richness index, which is the most straightforward index and is used to show species diversity in a location [44]; (2) the Shannon and Simpson index to measure community diversity; and (3) J Pielou index, which is an index commonly used to measure the evenness of species abundance [44,45]. These various measures of diversity were then calculated using Equations (1)–(4) in each sampling transect. Equation (1) is used to calculate the species richness index [46].

$$N = \text{number of species appeared in unit area,} \tag{1}$$

where N is the total number of species per transect sampling.

The Shannon (H) and Simpson (D) diversity indices were calculated using Equations (2) and (3), respectively [45,47,48].

$$H = \sum p_i \ln p_i \tag{2}$$

$$D = 1 - \sum p_i^2, \tag{3}$$

where $p_i$ is proportion of species in each transect sampling.

Pielou's J index was calculated using Equation (4) [44].

$$P = \frac{H}{\ln N}. \tag{4}$$

Evaluation of the performance of the two methods was analyzed by calculating the coefficient of determination ($R^2$) and its *p*-value. All analysis was carried out using the VEGAN package in R-4.1.0.

Next, from thermal camera data, we developed species detection within the study area based on supervised learning approaches: Random Forest, Support Vector Machine, and Boosted Regression Trees [49–51]. For this purpose, we specifically used DJI Matrices 200 series with 640 × 512 pixels' radiometric thermal camera (H20T—Quadsensor solution) and visible bands to detect species (i.e., Javan deer and water buffalo) within the

study area. We rescaled the images into 8-bit radiometric data (0–255) to create more efficient processes. We evaluated supervised learning using Random Forest and Support Vector Machine [49,52]. We used Random Forest algorithm with 1000 trees (N = 1000), then the number of variables per split has been defined as root square of the number of features. Moreover, Support Vector Machine (SVM) algorithm has been used to identify species as well as the other algorithms. We trained the data using One-class SVM with RBF kernel (i.e., radial basis function) [53]. Furthermore, we also created ensemble model from all algorithms based on principal component analysis (PCA; that is, performs a PCA with algorithms suitability and return the eigenvalues of the first component) [54].

We collected the training data of the species occurrences by visual interpretation from true color composite of drone imageries (Figure 3). To evaluate the models, we performed confusion matrix-based evaluation using several discrimination metrics—i.e., area under the curve (AUC; [55]); Kappa coefficient [56]; true skill statistic (TSS; [55]); Jaccard [57]; and Sorensen [57].

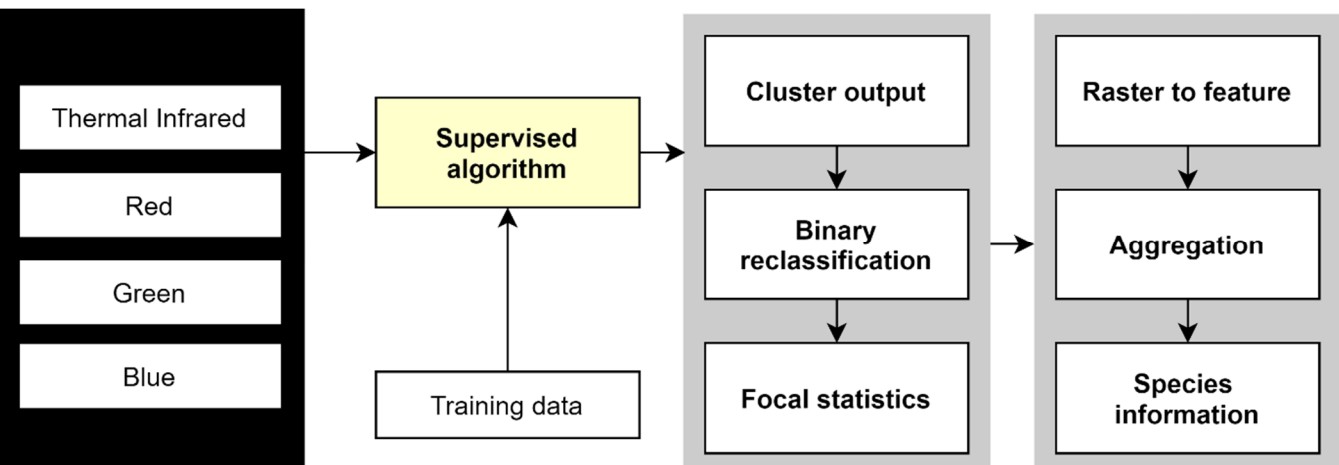

**Figure 3.** Methodological framework for species detection using multispectral and thermal of drone imageries.

2.4.3. Drone Flight and Behavioral Response of Primate in GHSNP

Two modelling frameworks were used to examine the differences between (1) detection and behavioral scores for each primate species based on ground and drone surveys, and (2) repeated behavior scores for each primate species based on a specific flight altitude. For that purpose, the behavior of each primate species was grouped and scored on a five-point scale (Table 1). We used the highest number score for each animal in each survey method (ground vs. drones) for statistical analysis during each observation period. The chi-square test was used to compare the resulting detection times with each method used. Furthermore, the Kruskal–Wallis and Dunn multiple comparison test was used to examine differences in behavioral responses of the three primate species in each survey method (ground vs. drones) [58]. The *p*-value used to account for several statistical tests was obtained through Benjamini–Hochberg analysis [59]. The generalized ordinal logistic mixed-effects model (a.k.a, proportional odds model) is used because behavioral scores at given altitudes are not normally distributed and are categorical (ordinal). In this model, the behavioral response as the dependent variable is applied and implemented with the ordinal package in R [60]. Behavioral scores between surveys are independent and no specific environmental covariates are used to compare ground-drone surveys.

**Table 1.** Ethograms used by observers to describe behavioral response scores (0 to 4) shown by target animals during monitoring activities.

| Score | Category | Description |
|:---:|:---:|:---|
| 0 | Resting | Behaviors include inactive (not moving), sleeping (e.g., closing eyes), and other behaviors such as grooming, playing, and breeding. |
| 1 | Vigilant | Noting around and alert with regular lateral movement of the head and eyes open. |
| 2 | Looking | Specifically spotting, tracking objects, and visually following sources of disturbance (an observer on the ground or drone flying in the vicinity) with the head possibly turned up or down or sideways left and right. |
| 3 | Agonistic | Aggressive behavior is shown to observers on the ground or drone flying in the vicinity, including alarm calls. |
| 4 | Escape | Move away from the source of the disturbance (an observer on the ground or drone flying in the vicinity), including hiding so that they are not detected again. |

## 3. Results

*3.1. Nest Counts and Observed Nesting Material and Breeding of Waterbirds in PRWS*

3.1.1. Interpretation of the Imagery

During ground and drone-based nest surveys, we identified a total of seven waterbird species from five families (Ardeidae, Phalacrocoracidae, Anhingidae, Threskiornitidae, and Ciconiidae). From the ground surveys, species included grey heron (*Ardea cinerea*), purple heron (*Ardea purpurea*), little black cormorant (*Phalacrocorax sulcirostris*), oriental darter (*Anhinga melanogaster*), little egret (*Egretta garzetta*), and black-crowned night heron (*Nycticorax nycticorax*). Nest monitoring using Matrice 300 RTK, identified 6 species; grey heron, purple heron, little black cormorant, milky stork, little egret, and black-crowned night heron. Nests can be seen and distinguished from the background in the RGB imagery and heat signature through the TIR imagery. Nests of various species of waterbirds were distinguished based on their dimensions (nest dimensions are commonly positively correlated with body size of waterbirds), nest locations in each colony of waterbird species (e.g., *Phalacrocorax sulcirostris* which is found with large individual numbers in the same location, or *Ardea cinerea* nests which are commonly located around and surrounded by *Nycticorax nycticorax* nests), and confirmed nest locations by heat markings at drone flight altitudes above 65 m consistently (e.g., *Ardea cinerea*, *Nycticorax nycticorax*, and *Ardea purpurea*; Figure 4). RGB imagery performance was better than heat signatures in the TIR imagery at the highest allowable flight altitude (90–120 m) in detecting nest presence but was limited to distinguishing nests of various recorded waterbird species (Figure 5). Finally, with RGB imagery, almost the entire nest can be seen even without adult birds or eggs/chicks in the nest, except for *Mycteria cinerea*, we failed to find their nests in the study area but recorded their activity on the ground in mangrove forest (Figure 6).

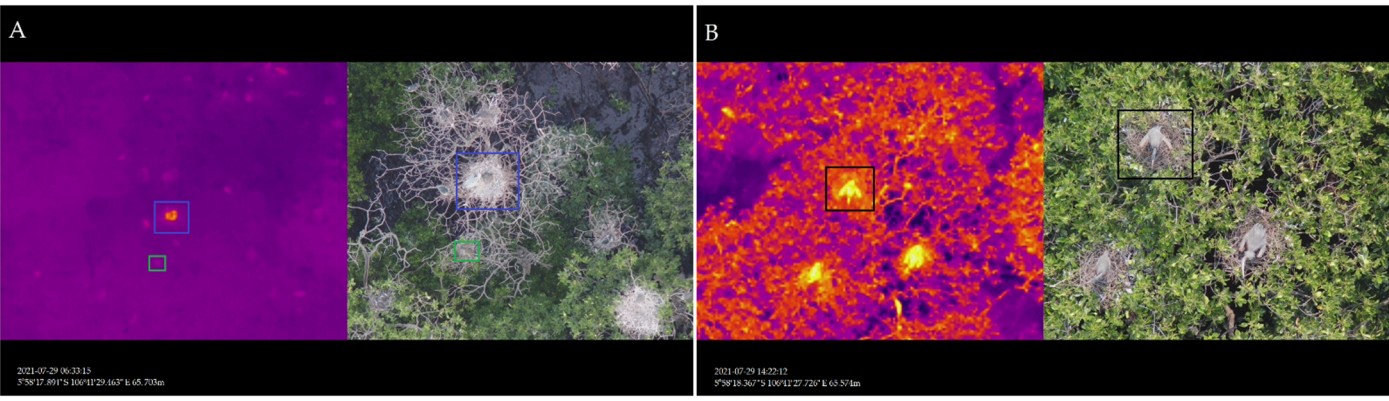

**Figure 4.** Nest on RGB imagery and heat signature through TIR imagery, (**A**) nest of *Ardea cinerea* (blue box) and nest of *Nycticorax nycticorax* (green box), and (**B**) nest of *Ardea purpurea* (black box). Nests with large dimensions show a higher temperature contrast between the object and its background.

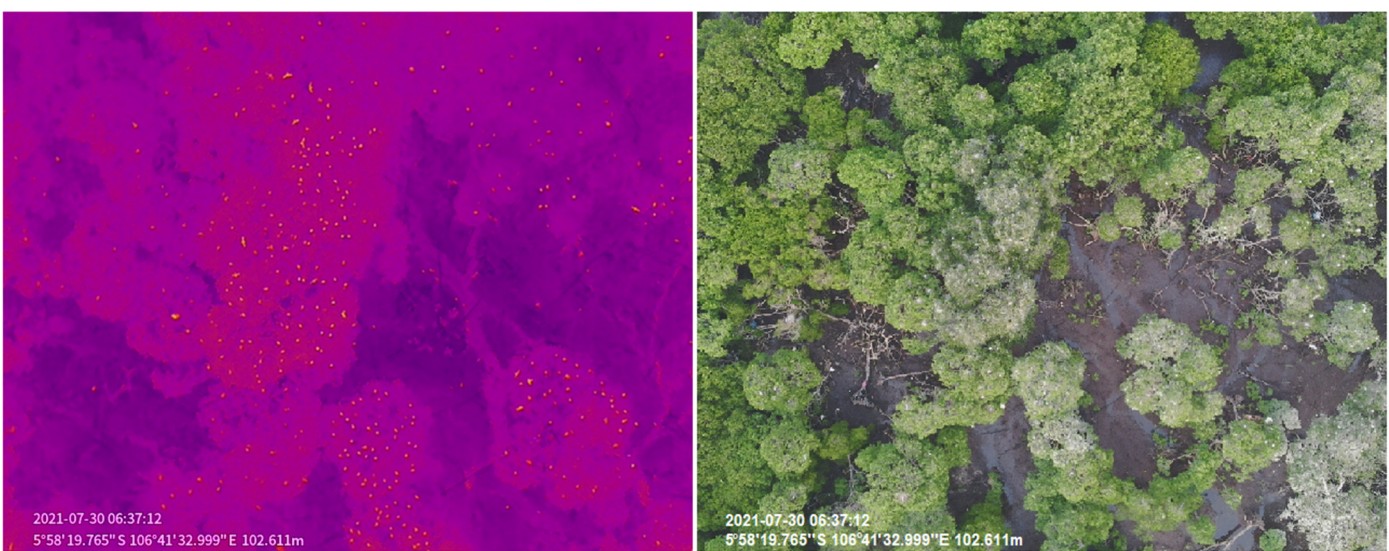

**Figure 5.** Nest on RGB imagery confirmed via TIR imagery on drone flights at an altitude of 100–120 m.

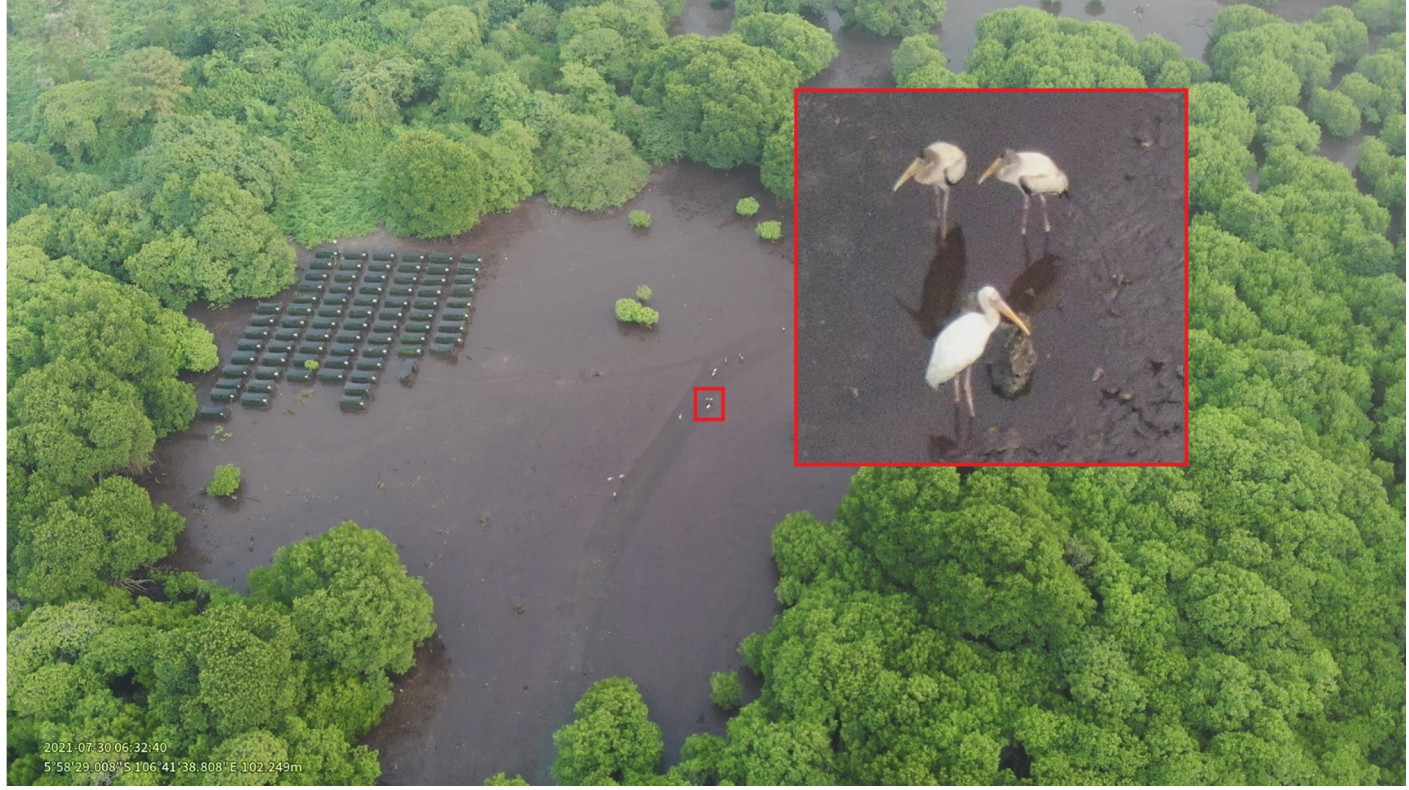

**Figure 6.** Waterbird species with the largest body size on the Pulau Rambut Wildlife Sanctuary with conservation status Endangered in the IUCN Red List.

### 3.1.2. Nest Number, Nesting Material and Waterbird Breeding Monitoring

A higher density of nests was detected from drone imagery compared to those observed during ground surveys (Table 2). Waterbird nests with small dimensions were difficult to distinguish from their background when using TIR imagery. Calculating waterbird nests through ground surveys is challenging since most nests are located at the top of the tree and covered by branches, twigs, or leaves underneath. Visual observations of 683 nests detected by drones found that 15.52% of waterbird nests were built using materials derived from marine litter including plastic (plastic rope; 84%), Styrofoam (8%), bamboo pieces (2%), and so forth (Figure 7). Four species (Ardea cinerea, Ardea purpurea, Phalacrocorax sulcirotris, and Nycticorax nycticorax) of the seven found during the survey were in breeding season, indicated by the presence of eggs and chicks. Of the 590 filled nests recorded, 74 nests contained eggs and 54 contained chicks (Figure 8).

**Table 2.** Nest number from ground and drone survey in Pulau Rambut Wildlife Sanctuary.

| Nest Category | Ground | Drone |
|---|---|---|
| Nest filled | 124 | 590 |
| Empty nest | | 93 |

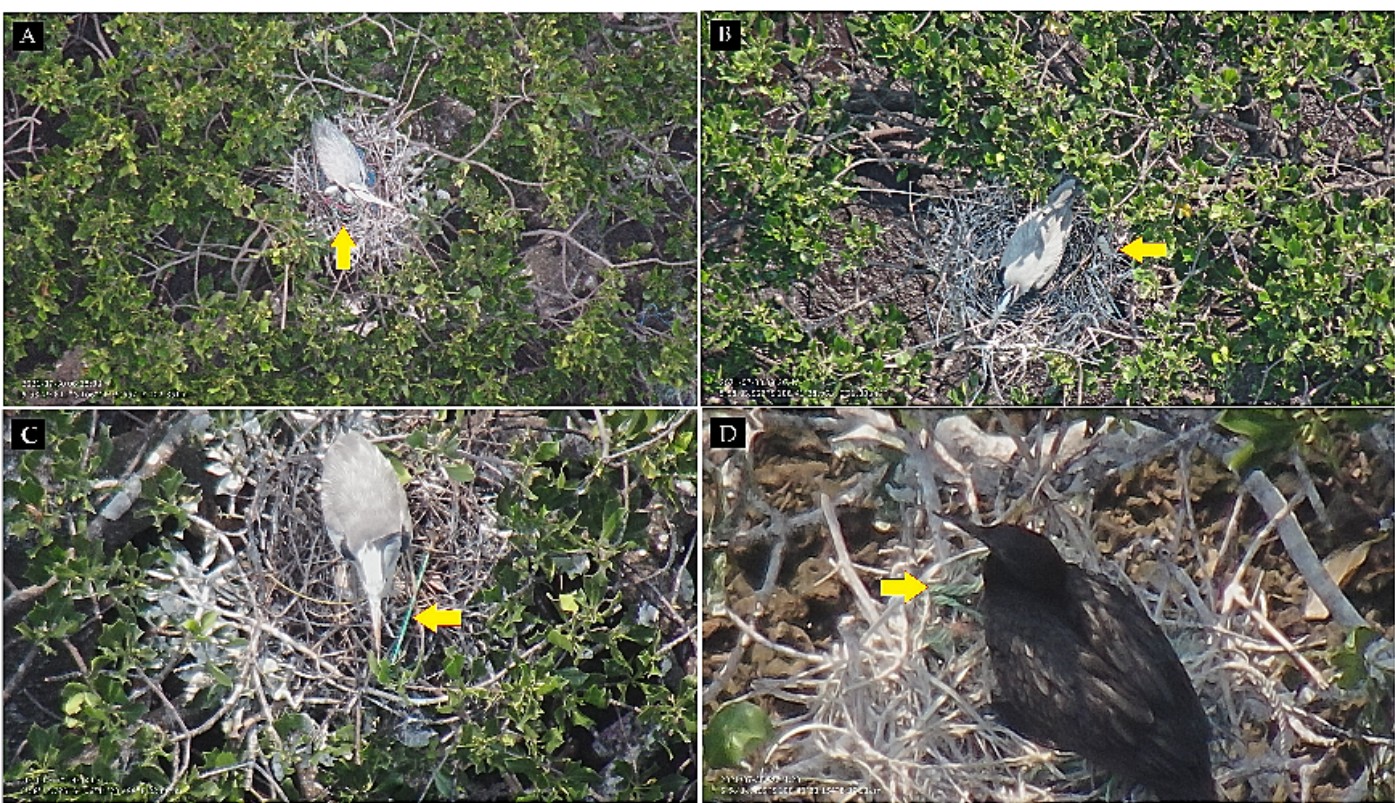

**Figure 7.** Marine litter used as nesting material: (**A**) plastic rope, (**B**) Styrofoam, and (**C**) straws in *Ardea cinerea* nests, and (**D**) plastic rope in *Phalacrocorax sulcirotris* nest.

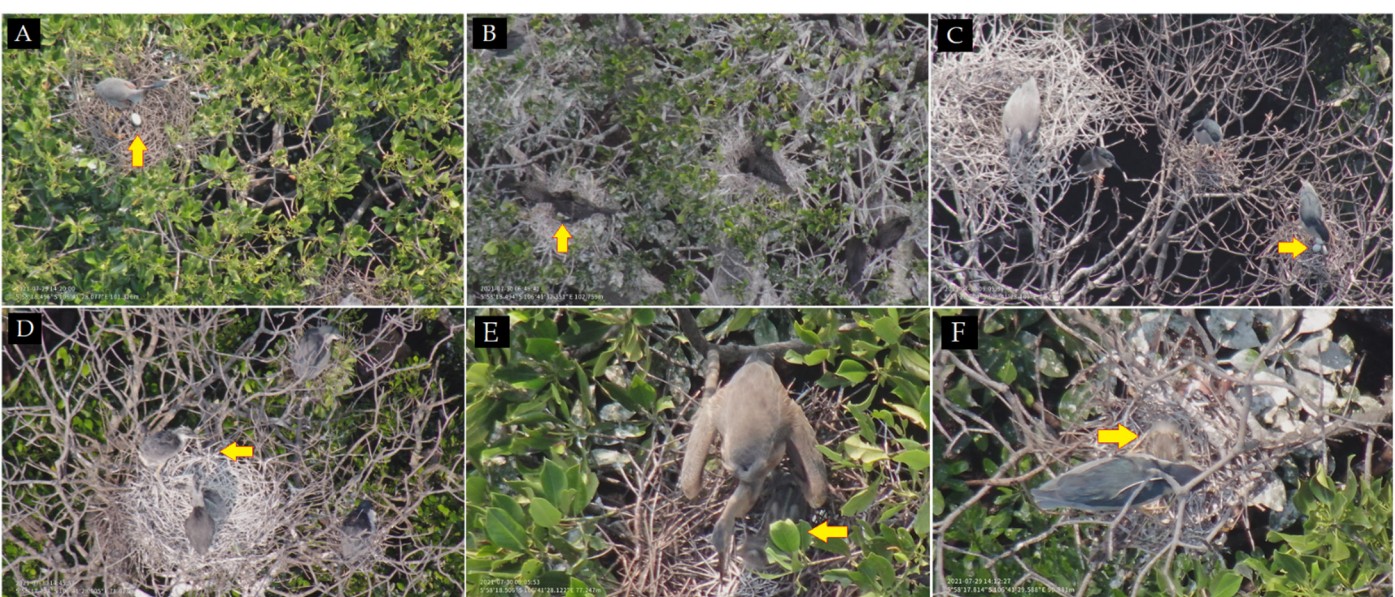

**Figure 8.** Eggs in waterbird species (**A**) *Ardea purpurea*, (**B**) *Phalacrocorax sulcirotris*, (**C**) *Nycticorax nycticorax*, and chicks in the species of (**D**) *Ardea cinerea*, (**E**) *Ardea purpurea*, (**F**) *Nycticorax nycticorax*.

### 3.2. Rapid Assessment of Species Diversity, Species Detection and Model Evaluation in BNP

#### 3.2.1. Relationship between Indices Estimated by the Ground and Drone Surveys

An earlier study employing camera traps (2019–2020), detected at least 20 species in the Savanna Bekol (Baluran National Park and Copenhagen Zoo, unpublished data). The mean percentage of species detected during our surveys was significantly larger for drone surveys than ground surveys ($p < 0.01$; Figure 9). Three species were not detected (Panthera pardus melas, Apodemus sylvaticus and Acridotheres tricolor), and one species (Acridotheres javanicus) was not recognized during our drone surveys. There was a significant linear relationship between the various indices expressing measures of species diversity (species richness, Shannon index, Simpson index and J Pielou index) estimated by ground surveys and drone surveys ($p < 0.01$, Figure 10). Drone surveys estimated the range of variation (VR), smaller than the ground survey method. Range of variation for species richness, Shannon index, Simpson index and Pielou's J index estimated by ground surveys was 4.45–15.45, 1.68–2.88, 0.78–0.94 and 0.70–0.88, respectively. Meanwhile, drone survey results were 13–17, 2.74–3.03, 0.90–0.95 and 0.87–0.90.

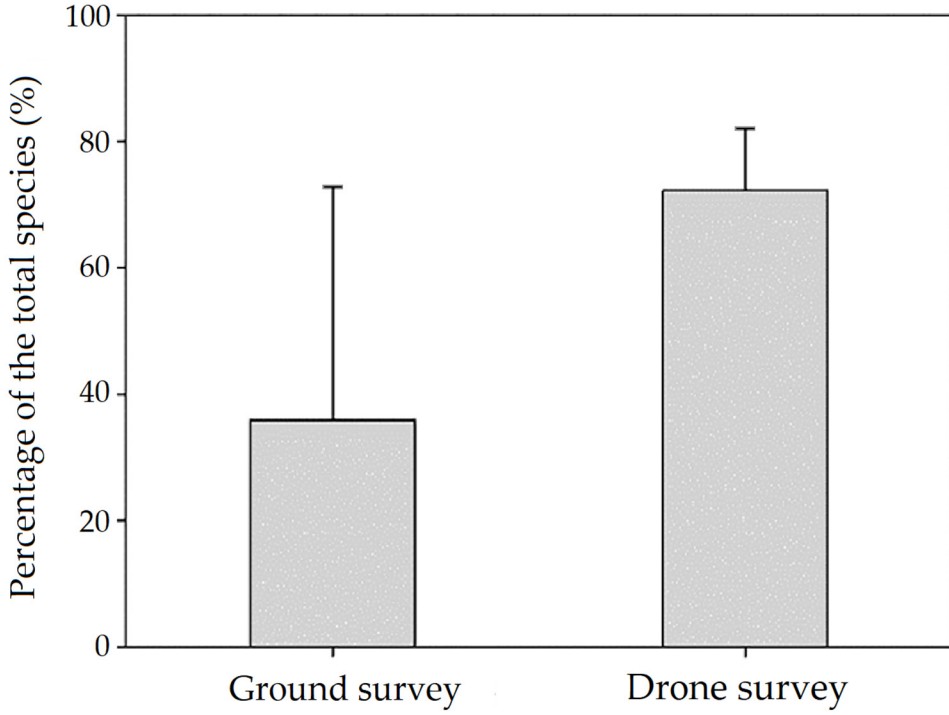

**Figure 9.** Percentage of total species from ground and drone surveys in sampling units in Savanna Bekol.

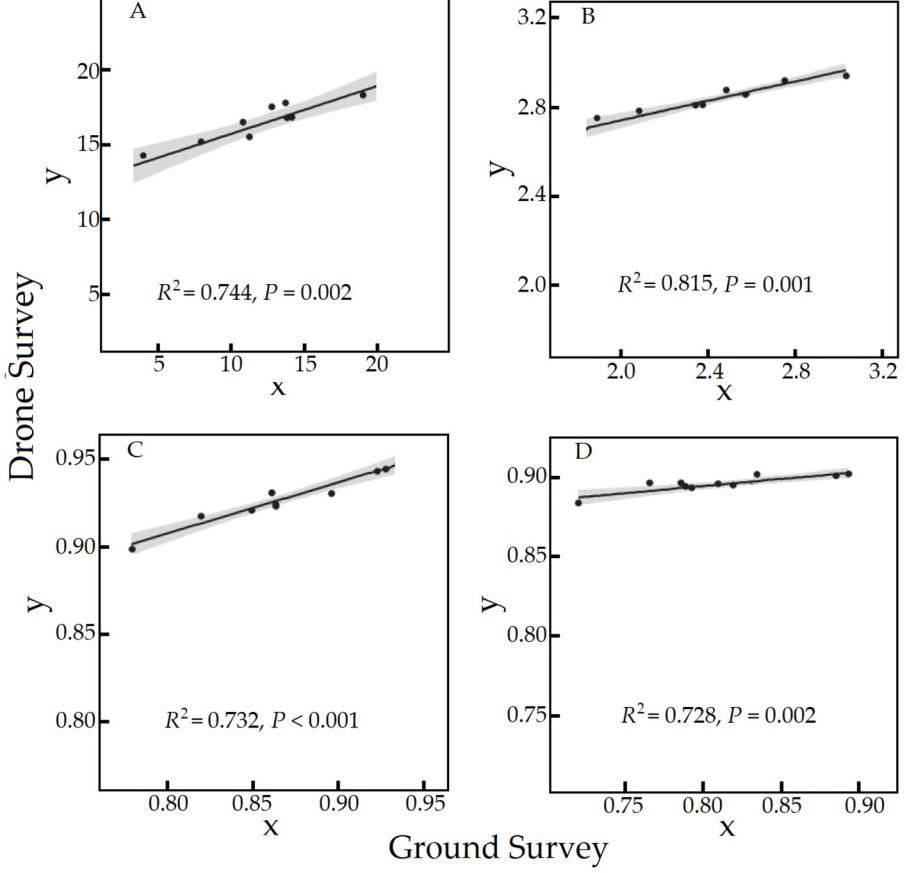

**Figure 10.** Estimation of the relationship between species richness (**A**), Shannon index (**B**), Simpson index (**C**) and J Pielou index (**D**) using ground survey and drone survey data.

### 3.2.2. Species Detection and Model Evaluation

We evaluated both Javan deer and water buffalo based on drone imagery (i.e., blue, green, red, and infrared thermal bands) using three different algorithms and an ensemble model. This study found that the ensemble model outperformed other algorithms for both species based on AUC, Kappa, TSS, Jaccard, and Sorensen. For Javan deer, the results show the overall means of 0.906 for AUC, 0.827 for Kappa, 0.827 for TSS, 0.852 for Jaccard, and 0.920 for Sorensen (Table 3, Figure 11). For water buffalo, the results show the overall means of 0.974 for AUC, 0.951 for Kappa, 0.951 for TSS, 0.951 for Jaccard, and 0.978 for Sorensen (Table 3, Figure 12).

**Table 3.** Model evaluation using different discrimination metrics for Javan deer and water buffalo detections.

| Species | Algorithm | AUC | Kappa | TSS | Jaccard | Sorensen |
|---|---|---|---|---|---|---|
| Javan deer | BRT | 0.901 | 0.827 | 0.827 | 0.852 | 0.920 |
| | SVM | 0.895 | 0.827 | 0.827 | 0.852 | 0.920 |
| | Random Forest | 0.893 | 0.827 | 0.827 | 0.852 | 0.920 |
| | Ensemble | 0.936 | 0.827 | 0.827 | 0.852 | 0.920 |
| Water buffalo | BRT | 0.979 | 0.944 | 0.944 | 0.956 | 0.975 |
| | SVM | 0.958 | 0.944 | 0.944 | 0.956 | 0.975 |
| | Random Forest | 0.965 | 0.944 | 0.944 | 0.956 | 0.975 |
| | Ensemble | 0.993 | 0.972 | 0.972 | 0.978 | 0.988 |

Our results show that the ensemble model can detect the object of both species with relatively good performance. We found eight Javan deer individuals and 33 individuals of water buffalo. We found that the ensemble model also can reduce speckle noise of the model output that creates a spatial bias for the model. However, we also found patches of errors from Javan deer images due to burn scars in the non-vegetated areas.

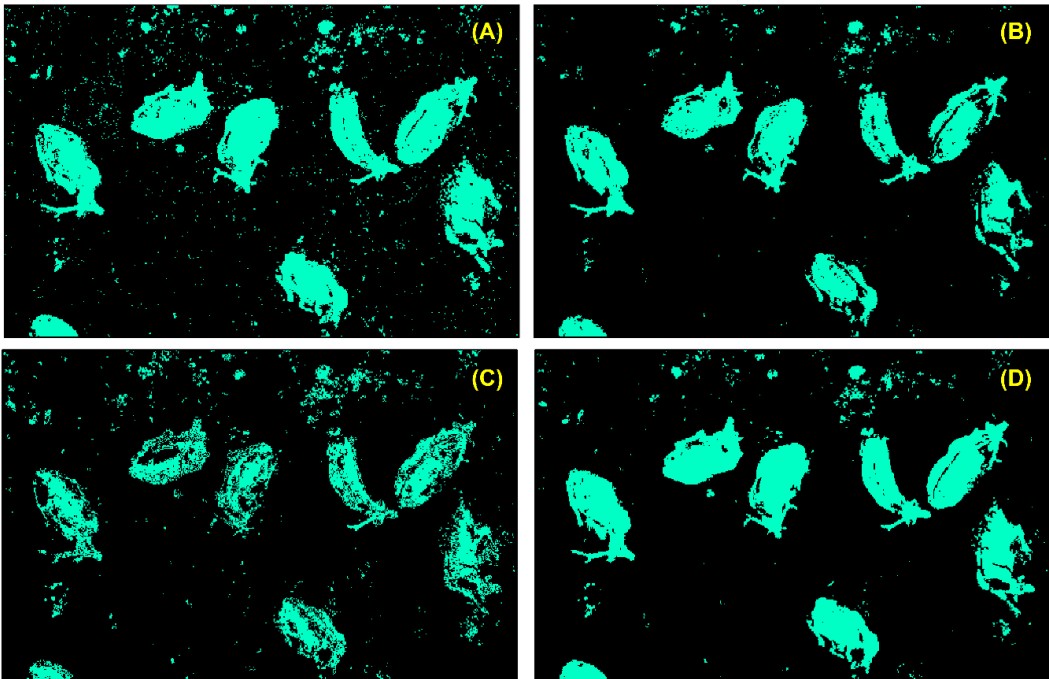

**Figure 11.** Javan deer detections based on drone imagery using machine learning approaches: (**A**) Support Vector Machine, (**B**) Random Forest, (**C**) Boosted Regression Trees, and (**D**) Ensemble model.

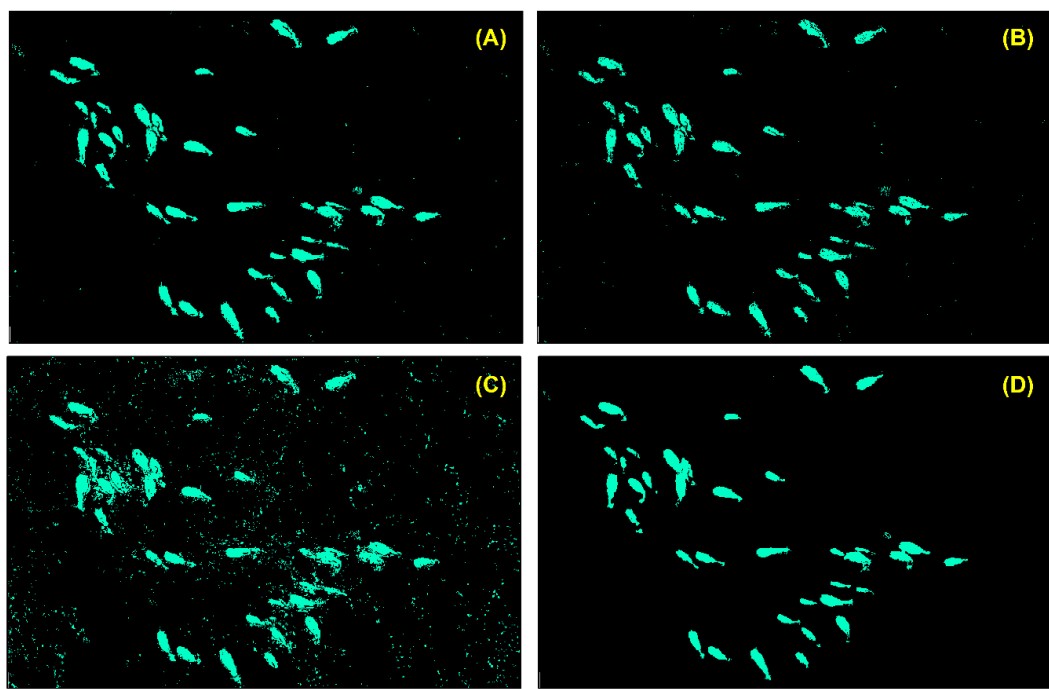

**Figure 12.** Water buffalo detections based on drone imagery using machine learning approaches. (**A**) Support Vector Machine, (**B**) Random Forest, (**C**) Boosted Regression Trees, and (**D**) Ensemble model.

This study found that thermal imagery from drones had relatively significant performance to detect species (i.e., Javan deer or water buffalo) (Wilcoxon test: Water buffalo, W = 2364985, *p*-value < 0.005; Javan deer, W = 7455116, *p*-value < 0.005). Javan deer temperature ranging from 26.3 °C to 43.7 °C ($\bar{x}$ = 33.53 °C; s.d. = 3.92 °C) and water buffalo temperature ranging from 25.1 °C to 39.0 °C ($\bar{x}$ = 31.09 °C; s.d. = 2.99 °C) (Figure 13). Our used thermal sensors had relatively low bias (±1 °C) and acceptable to capture temperature body for mammal species.

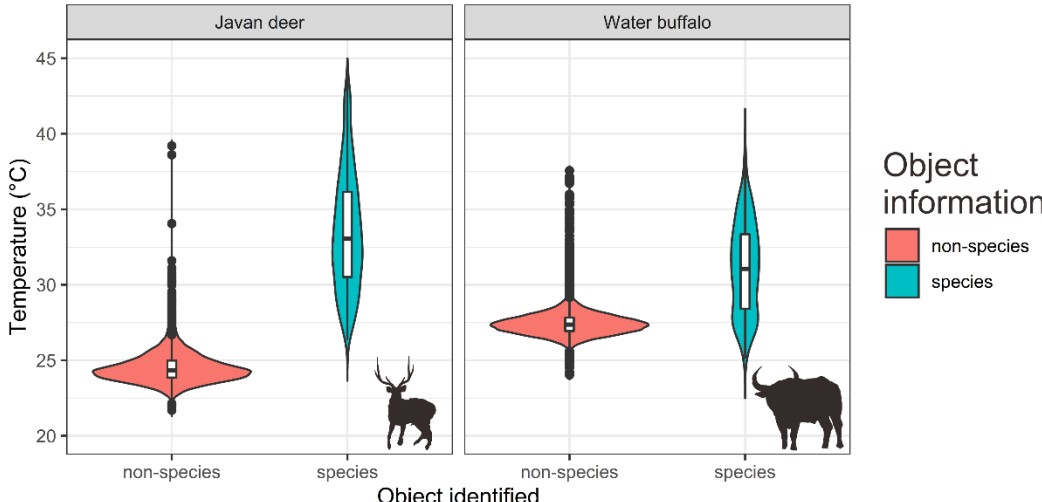

**Figure 13.** Temperature responses from species body for Javan deer and water buffalo in the Baluran National Park.

*3.3. Behavioral Scores of Three Primates in GHSNP*

3.3.1. Survey Comparison Study

We flew 36 drone flight missions (including 42 target group approaches) and conducted 15 ground surveys focused on three groups of Javan gibbon, eight groups of Javan langur, and three groups of Javan surili. Average group sizes were 4.5 (s.d. 1.2) for Javan gibbon, 5.4 (s.d. 3.7) for Javan langur, and 3.4 (s.d. 1.8) for Javan surili. The total sample sizes and number of behavioral scores assigned are listed in Table 4. For all three species behavioral response scores differ significantly between drone surveys (30 m altitude) and ground survey, for Javan langur and Javan surili (Table 5 and Figure 14).

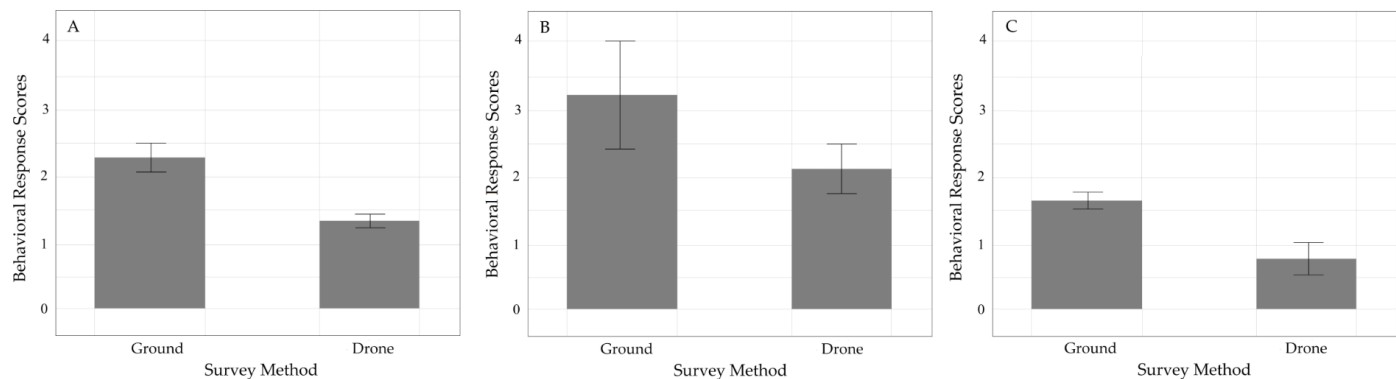

**Figure 14.** Average behavioral responses (plotted with error bars representing the standard errors): (**A**) of Javan gibbon (*n* = 138 scores), (**B**) Javan langur (*n* = 432 scores) and (**C**) Javan surili (*n* = 198 scores) to drones overflights at 30 m versus ground surveys.

**Table 4.** Numbers of flights, ground surveys, and individual animals scored by independent observers for behavioral reaction to drones approach listed by species and study.

| Species | Comparison Study | Number of Individuals | Number Drone Approaches | Number Ground Survey | Number of Behavioural Scores per Observer | Total Number of Behavioural Scores |
|---|---|---|---|---|---|---|
| Javan gibbon | Survey | 7 | 6 | 6 | 46 | 138 |
| | Altitude | 12 | 8 | - | 102 | 306 |
| Javan langur | Survey | 10 | 16 | 18 | 144 | 432 |
| | Altitude | 25 | 38 | - | 268 | 804 |
| Javan surili | Survey | 7 | 6 | 6 | 66 | 198 |
| | Altitude | 23 | 10 | - | 236 | 708 |

**Table 5.** Comparison of the behavioral responses of three primates grouped by type of survey using the Kruskal-Wallis test.

| Species | *n* | Treatment | df | χ2 | *p* |
|---|---|---|---|---|---|
| Javan gibbon | 46 | All surveys | 2 | 4.58 | <0.050 |
| | 34 | Ground-drone | | | <0.050 |
| Javan langur | 144 | All surveys | 2 | 38.50 | <<0.001 |
| | 100 | Ground-drone | | | <<0.001 |
| Javan surili | 66 | All surveys | 2 | 24.78 | <<0.001 |
| | 40 | Ground-drone | | | <<0.001 |

### 3.3.2. Altitude Comparison Study

The increase in the behavioral response of Javan gibbon and Javan surili occurred at an altitude of <20 m, while for Javan langur occurred at an altitude of <30 m. Behavioral responses gradually decreased and were indistinguishable at flying altitudes > 20 m for the Javan gibbon and Javan surili populations, while Javan langur > 30 m (Figure 15). An increase in behavioral responses such as escape (4) is common during low flight and decreases to alert (1) and rest (0) with an increase in altitude above 20 m. The ordinal logistic model fit the data well, and the altitude effect provided an informative model for the three primate species (Hess condition score < $10^3$, Table 6). The value for <20 m is positive, indicating an increase in the behavioral score when the UAV flies lower. Meanwhile, behavioral reactions at heights > 20 m did not inform any model (Table 6). Behavioral reaction scores increased after incorporating the effect of sex and altitude when Javan langurs were resting and guarding their cub ($\chi2$ (1) = 27.50, $p \ll 0.0001$). The most informative model for Javan gibbon ($\chi2$ = 1.50, $p \ll 0.0001$) and Javan surili ($\chi2$ = 14.90, $p < 0.0001$) included effects for altitude, and human exposure level. Males and animals with low previous exposure to humans were more likely to have higher scores than females and high exposure animals, respectively. Observer as a random effect was significantly informative to every model framework in which it was tested (paired likelihood ratio test $p < 0.05$, Table 6).

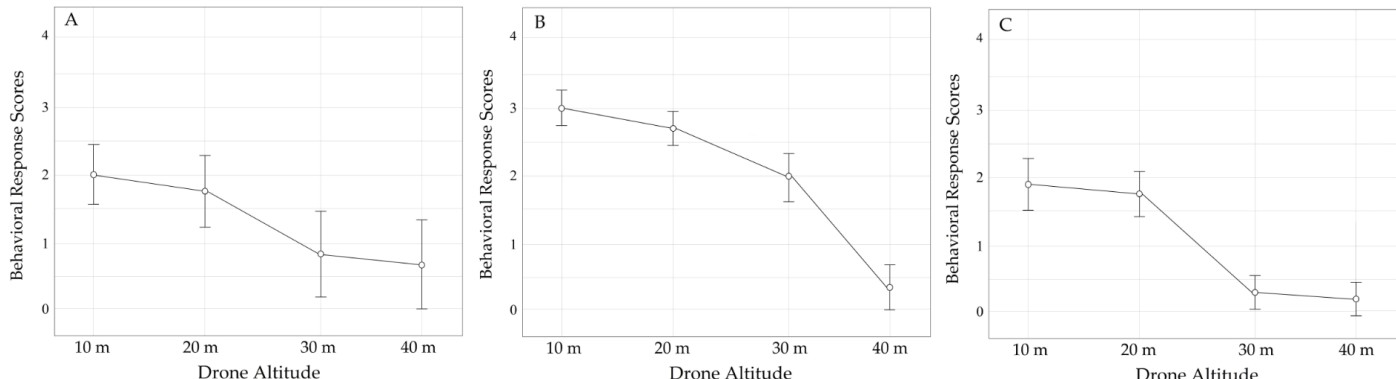

**Figure 15.** Average behavioral responses (plotted with error bars representing the standard errors): (**A**) of Javan gibbon (*n* = 46 scores), (**B**) Javan langur (*n* = 144 scores) and (**C**) Javan surili (*n* = 66 scores) to drones overflights at 30 m versus ground-based surveys.

**Table 6.** Generalized ordinal logistic mixed-effects model results from the behavioral response scores of three primates species during the observation period from January to March 2021. Altitude (levels = 30 m, 20 m, 10 m), Social Group (levels = Harem, Non-harem), Sex (levels = Male, Female), and Exposure (High, Low) were fixed effects, and Observer (levels = Obs1, Obs2, Obs3) was a random effect. Likelihood ratio test *p*-values reflect comparison with the model in the row above. ΔAIC values indicate the difference from the best-performing model = bold text. The intercepts were significant in all models (Javan gibbon $\beta_0$ range: −0.368 to 4.788, $p \ll 0.001$; Javan langur $\beta_0$ range: −0.208 to 5.290, $p \ll 0.001$; Javan surili $\beta_0$ range: −0.054 to 5.864, $p \ll 0.001$). β model coefficients back-transformed from log space.

| Model | Significant Coefficients | | AIC | Δ AIC | Likelihood Ratio Test | |
|---|---|---|---|---|---|---|
| | β | Pr(>\|z\|) | | | (Df) χ² | *p* |
| Javan gibbon | | | | | | |
| Response~1 + (1\|Observer) | - | - | - | 964.20 | 88.80 | - | - |
| Response~Altitude + (1\|Observer) | 10 m | 1.690 | <<0.001 | 910.70 | 44.40 | (1) 35.44 | <<0.0001 |
| | 20 m | 0.978 | <<0.001 | | | | |
| **Response~Altitude + Sex + Exposure + (1\|Observer)** | 10 m | 1.554 | <<0.001 | 840.50 | - | (2) 1.50 | <<0.0001 |
| | 20 m | 1.158 | <<0.001 | | | | |
| | Up | 1.088 | <0.001 | | | | |

| | | | | | | | |
|---|---|---|---|---|---|---|---|
| | Male | 0.645 | 0.001 | | | | |
| | Low | 0.630 | <0.001 | | | | |
| Javan langur | | | | | | | |
| Response~1 + (1|Observer) | - | - | - | 5268.20 | - | - | - |
| Response~Altitude + (1|Observer) | 10 m | 2.764 | <<0.001 | 3445.10 | 924.50 | (4) 158.22 | <<0.0001 |
| | 20 m | 1.890 | <<0.001 | | | | |
| Response~Altitude + Social Group + (1|Observer) | 10 m | 2.893 | <<0.001 | 2914.80 | 278.90 | (2) 102.40 | <<0.0001 |
| | 20 m | 1.797 | <<0.001 | | | | |
| | Harem | 0.814 | <0.001 | | | | |
| **Response~Altitude ∗ Social Group + Sex + (1|Observer)** | 10 m | 0.690 | <<0.001 | 2544.70 | - | (1) 27.50 | <<0.0001 |
| | 20 m | 0.758 | <<0.001 | | | | |
| | 10 m: Harem | 2.427 | <0.001 | | | | |
| | 20 m: Harem | 1.564 | <0.001 | | | | |
| | Male | 0.261 | <0.001 | | | | |
| Javan surili | | | | | | | |
| Response~1 + (1|Observer) | - | - | - | 7540.80 | - | - | - |
| Response~Altitude + (1|Observer) | 10 m | 1.664 | <0.001 | 7288.10 | 1556.40 | (4) 358.10 | <<0.0001 |
| | 20 m | 0.981 | <0.001 | | | | |
| **Response~Altitude + Social Group + Sex + Exposure + (1|Observer)** | 10 m | 1.603 | <<0.001 | 5580.60 | - | (1) 14.90 | <<0.0001 |
| | 20 m | 0.965 | <<0.001 | | | | |
| | Harem | 0.885 | <0.001 | | | | |
| | Up | 0.253 | 0.001 | | | | |
| | Male | 0.500 | <<0.001 | | | | |
| | Low | 1.610 | <0.001 | | | | |

## 4. Discussion

### 4.1. Nest Counts and Observed Nesting Material and Breeding of Waterbirds in PRWS

Drone surveys produced high-resolution images of colonies of several species of waterbird at PRWS, both resident and non-resident species during the survey. The number of species and nests found in our survey was not as high as in previous surveys (see: [37,39]). There was a high variation of species and numbers throughout the year in PRWS, with the greatest abundance usually occurring during June-July and the lowest abundance in November-December, coinciding with the worst West Monsoon season [37]. Seasonal resident species came to the island to breed, and soon after, their young were able to fly and leave the island, for example, *Mycteria cinerea*, *Plegadis falcinellus*, *Threskiornis melanocephalus*, and *Bubulcus ibis*; [37]. From orthomosaic, we determined the nest formed, its spatial location, population growth, and threats to the conservation of the waterbird population in this location. The spatial resolution achieved by flight allows us to perform well-breeding population estimates, since nest structures and individual waterbirds can be identified in orthomosaic. Manual calculations on each grid with full of caution using visual imagery and confirmed by thermal imagery consistently successfully detect nests and waterbirds in the image. Our drone surveys produce a higher number of nests in comparison to traditional ground surveys. Furthermore, the advantage of using drones is that they allow a better perspective from a wider and higher vantage point, especially when several nests are above vegetation or among dense vegetation, which is likely to contribute to low detection and higher miscalculation in ground surveys. In line with other comparisons between survey methods, including counts of nesting shorebirds colony [61] and seabird nesting [62], drone surveys result in more accurate counts than ground surveys.

In a relatively short study period, the results of our research through initial monitoring and orthomosaic provide various information related to the distribution of waterbird

nests in PRWS, being: (a) there are five waterbird colony locations on PRWS; (b) the south-eastern part of the island is intensively used all year round; and (c) some areas that are never used for nesting by waterbirds, namely coastal forests on the south and west sides of the island. This is in line with previous research that was conducted intensively through ground surveys [37]. Furthermore, we found positive and negative associations between species pairs based on their nesting sites. There is a highly positive and significant relationship, for example, between *Ardea cinerea* and *Nycticorax nycticorax*. In contrast, negative associations were found between congeneric species (i.e., *Ardea alba-Egretta garzetta*).

The waterbird breeding census at PRWS using ground surveys is challenging. Nests are built on macrophytic vegetation over two meters high, with colonies housing many individuals of each species. To detect the presence and count the number of eggs and chicks in the nest by climbing each nest tree requires time, effort, and poses safety risks. The results obtained from this study demonstrate the capability and benefits of drone surveys for detailed monitoring of all water birds that inhabit the island during the monitoring period, and its applicability can be used to provide an overview of long-term waterbird population trends. Drones can collect high-resolution spatial data over large and inaccessible areas at an affordable cost depending on the surface to be covered [63,64]. Another benefit is that in its application, drone flights at low altitudes do not provide significant disturbance to waterbirds even when they are breeding. In situations where the waterbird population is increasing, monitoring using traditional methods is not feasible. The resolution achieved in our monitoring activities allows for a manual approach to nest counting, but in the future, there will be a need for automated estimates that simplify, shorten data processing time and increase the precision of calculation results. As drone capabilities will improve over time, particularly concerning flight endurance and sensor resolution, large amounts of high-resolution data will be collected, and automated processing will become a major requirement in the future [65]. Adapting automated procedures is critical for incorporating drone data into long-term monitoring programs, as this will reduce bias due to visual interpretation and provide comparable data over time.

The impact of marine litter on the waterbird habitat can be seen from the overlap between marine litter and trees that become waterbird nests. Some of the impacts include: (1) marine litter that accumulates to cover 50% of mangrove roots can cause depressed mangroves, which is marked by leaf loss, while marine debris that covers 100% of mangrove roots will cause the death of mangrove stands that are used as nest trees [66]; (2) the use of synthetic materials such as rope can harm birds; this can be seen in the case of a *Phalacrocorax sulcirostris*, which died entangled in rope in its nest; (3) as many as 51 trillion metric tons of microplastics were found in the sea [67] and have entered the food chain [68]; it is not impossible that microplastics and macro-sized plastics found in nests are swallowed by water birds and can cause death [69].

*4.2. Species Diversity and Detection Techniques from Thermal Photographs of Two Large-Sized Mammals in Baluran National Park (BNP)*

This study is the first to be conducted in a savanna ecosystem in tropical areas, using drones to monitor the diversity of resident wildlife. The results provide basic information that can be used for future wildlife monitoring across a wide and open area. Drone technology appears to be a suitable method for rapidly collecting data on the diversity of wildlife species in this habitat due to its ability to take high-resolution images which can be accurately identified in the laboratory and compared over time and reduce the uncertainty of estimates in traditional observer calculations [70].

In applying the survey method, it is crucial to evaluate the survey results and the level of disturbance to the animals being surveyed. In our research, the detection rate of wildlife species diversity is not much different from the use of camera traps that are proven reliable and rigorous in detecting terrestrial wildlife. Although carried out in a relatively short period, drones have captured the presence of more than 80% of animals commonly found in this savanna area. The drone results also indicate a stark difference

with the ground survey method, which gives poor results. The high level of disturbance to animals during ground surveys affects the results provided through this survey activity. Drones show unique advantages in animal diversity studies, this platform can directly monitor data and animal distribution accurate and in real-time, primarily surveys on the population numbers of large animals and animal conservation. Our study observed no behavioral changes that could be interpreted as a nuisance in wildlife observed using drones. Although it is believed that periods of non-breeding and animals in large groups are shown to trigger behavioral reactions [71], we did not find this in our study. Various types of animals are not seen moving in successive photos. This indicates that flying at altitudes above 50 m seems appropriate and provides adequate resolution for animal identification. However, it is essential to note that disturbances are not always expressed in behavioral responses but can also occur in the form of physiological responses, such as changes in heart rate [72]. In our study, we did not specifically monitor the physiological responses of the animals being observed.

Previous studies have shown that machine learning provides good species detection performance [73–75], and our study concurred that using machine learning approaches, notably thermal and RGB imageries, detected both Javan deer and water buffalo populations as experimental species. We found that water buffalo detectability performed better than for Javan deer. The resting activity of Javan deer created thermal bias between body temperature and surface or environmental temperature. Therefore, resting Javan deer provided a lower detectability than water buffalo. Our study highlights the need to understand activity time budgets of the focal species prior to assessing the population through drone surveys.

Our study also showed that thermal information through object temperature provides good species and non-species detection results, particularly for warm-blooded species. We found that thermal drones can significantly differentiate the species from its environment or habitat. Javan deer have a higher body temperature than water buffalo. A previous study also showed the successful use of thermal imageries for wildlife species detection [76,77].

This study has revealed that using thermal drones in species detection for wildlife conservation based on machine learning approaches is promising due to its usefulness in generating high-quality data for the species population. Many conventional approaches for ecological surveys are challenging—that is, high costs and efforts, considerable time, high-level observer skills required and inaccessibility to rugged areas [78]. Thermal imagery along with visual RGB bands provides high-quality predictors in detecting species in their habitats. However, wider coverage areas of drones should be performed to obtain more various species and habitats representativeness.

*4.3. Behavioral Response of the Three Primate Species in Gunung Halimun Salak National Park (GHSNP) to the Flight of UAVs*

Although there are substantial differences in the life history, body size, group number, social behavior and ecology of each of the focal primate species in this study, the assessment of the behavioral responses to the presence of small drones flying at certain altitudes provides a similar pattern. Behavioral response to drone flights at higher altitudes is limited but increases as the drone's flying height decreases below 30 m. For Javan gibbons, eight groups inhabit the CRRF area in GHSNP of which three groups were habituated to human presence. Habituated and unhabituated groups of Javan gibbons affected the behavioral response of this species when observed using drones. The difference in behavioral responses is highly significant, with no response from habituated Javan gibbon when the drone approached up to 8 m. When the drone approached to a distance of 7 m, the Javan gibbon began to move slowly; however, habituation to regular human exposure appears to decrease sensitivity to drone use. In unhabituated Javan gibbons, which were the main target of this study, we observed that individuals did not respond when the drone was flying at 30 m and started moving away when the drone was lowered to 27

m. The Javan gibbon is a cryptic animal and tends to avoid the presence of humans [79], the behavioral response of the unhabituated Javan gibbon demonstrates the natural behavior of this species when it feels disturbed and threatened. For identification, the results from the RGB imagery were used since the thermal imagery was not possible even when flown at a lower altitude.

Javan langurs were more sensitive to 30 m overflights than Javan gibbon, and males were more likely to react than females. Behavioral responses also intensified over the course of the infant birth period. The Javan langur has a polygynous social structure where a single male controls territory and actively retains females within that space. As a result, female reactions to external stimuli frequently initiate a chain reaction where the male rushes through the harem to curtail female movement, impacting and disturbing other animals along the way. Vigilance is one of the functions of living in groups [80,81] because each group has the same role in detecting disturbances around them. If the disturbance is detected by one of the individuals in the group, the information will be passed on to other group members. Hence, disturbances are often amplified. Finally, the detection of drone noise is a primary source of animal disturbance across animal taxa [71,82]. We found Javan surili were significantly more likely to react when approached from upwind likely because drone noise is more strongly propagated to target animals during upwind approaches. Therefore, we suggest that behavioral reactions would be lower if groups are approached from down-wind.

The detection and monitoring of Javan surili using drones have challenges related to vertical space use in trees. Javan surili is most often found in trees at heights of 5–10 m, and some are also found at 4–18 m. Compared with the Javan gibbon, which uses trees at heights between 11–25 m, and Javan langurs (15–20 m). It is difficult to detect and observe Javan surili using RGB imagery when they use space in the lower altitude in trees. The thermal camera is effective in detecting the movement of the Javan surili under a tree canopy. The thermal camera will record any sign of the Javan surili body temperature that contrasts with the background. When drones fly over them, Javan surili has a limited behavioral response compared to Javan gibbons and Javan langurs. The closest altitude the drone can reach is at 23 m, and behavioral response is limited at this altitude. We believe this is related to the pattern of space use in the trees selected by the Javan surili. By being under the canopy, interference caused by drones can be minimized. Furthermore, being under the canopy can protect against aerial predators.

It should be noted that behavioral response studies are determined mainly by subjective assessments of the responses showed by animals. Behavioral studies almost exclusively still rely on a single observer, leading to unknown biases in their measurements. Although there are possible similarities in inter-observer scoring in behavioral response studies, information on inter-observer variability is measured and tested in appropriate mathematical models. Using two or three independent observers allows an assessment of the observer effect, and the results can reduce the possibility of bias due to observer subjectivity. The use of drones capable of digitally recording the target animal in images or videos allows multiple observers to view permanent records to ensure reproducibility.

## 5. Conclusions

Drones are increasingly being used to survey forest vertebrates, although commonly, most surveys are conducted to monitor primates, birds, and other species living and/or nesting in the canopy [83–85]. Due to the high and wide point of view, in our study, drones overcame some of the complex habitat-related visibility problems of surveying focal animals inhabiting tropical forests. Low visibility in the field due to the presence of vegetation and habitat complexity is a major source of bias in species monitoring activities in tropical forests. Our study detected more focal species in each survey area using drones than traditional ground surveys despite vegetation covering varied in coastal ecosystems, mangroves, savannas, and montane forests. We saw that drones effectively allow observers to see a wider range of spaces that would normally be difficult to observe, in this case,

beyond the first layer of vegetation to the tree canopy, and more generally able to reach inaccessible or distant and remote areas [86].

Drones are an approach to monitoring animals in a rigorous and non-invasive way to characterize the observed animals while also leaving a permanent record of their survey activities. Moreover, recorded images, videos or maps from drones can be used to verify species and their positions, numbers and sizes, including the occupied habitats of all detected species. Reducing disturbance to animals compared to ground surveys is also likely to result in more accurate data and avoid double-counting individuals as they escape and reappear elsewhere in response to disturbance. In our study, a lower behavioral response to external disturbances allows more intensive observation of animal behavior to be carried out. The use of drones has long been recognized as an advantage in aerial surveys of various animals [87,88]. The latter is a major problem for ecosystems in tropical forests where some animals will always be under tree cover and, therefore, undetected from above. In Indonesia's tropical forests, this means that drone surveys may not be a suitable method for monitoring various species of terrestrial animals that prefer forested habitats and daily activities under closed canopy forest cover, unless as technology advances in the future, commercial drones are available with robust sensors capable of penetrating the high-density vegetation commonly found in tropical forests. In our study, for example, we might miss some waterbird nests under the tree canopy. Although the numbers are small, of course, this is a bias factor for the accuracy of estimating the number of waterbird nests in PRWS. Our drone survey also failed to detect one of the endemic primate species present in the GHSNP, namely *Nycticebus java*nicus. This is likely to have contributed to its rarity as well as detection problems with ground surveys, as our nocturnal surveys also failed to detect them.

Despite all the advantages, drone monitoring still has some limitations. For example, the load carried by drones is low, and it is difficult to integrate multiple sensors in the same platform for observation, which is currently still based on optical cameras. Due to limited battery capacity, drones can only stay in the air for a short time and thus cannot carry out long-term wildlife monitoring. The limited battery life of the drone may limit the size of the study area surveyed, including constraints on remote study sites where access to electricity to recharge batteries is difficult and may limit the choice of study sites. We also found that strict national laws in each country and regulations in protected areas could prevent some further exploration of the use of drones to find the most effective and efficient approach to wildlife monitoring activities in tropical forests. In addition, the monitored area is smaller than other remote sensing platforms. In contrast to areas with unobstructed views, such as grasslands or savanna, it is difficult to get species information directly in the forest [62]. While there are other drawbacks to animal detection or signs from wildlife under the canopy, drone surveys tend to bring several advantages in multi-species research that inhabits diverse ecosystems in tropical forest areas. First, due to the high-resolution images of each survey activity in the three ecosystems (PRWS: 2.24 cm per pixel at 65 m; BNP: 1.72 cm per pixel at 50 m; GHSNP: 1.38 cm per pixel at 45 m), drones better detect animals and their signs in open areas or above tree canopies. While this is possible through close individual approaches during ground surveys, even expert observers have proven error-prone, and close approaches may be stressful for animals. Images or videos produced from drone surveys can also identify the various species present at the study site. Finally, we believe that drones are a reliable tool and will continue to increase their use for multi-species surveys in tropical forests. We observe that drones are becoming less expensive over time and are easier to use in their application. These tools can often facilitate field logistics and reduce costs compared to traditional ground survey methods, requiring multiple observers with strong field experience.

**Author Contributions:** D.A.R. conceptualized and designed the experiments; A.B.Y.S. performed the experiments with guidance from D.A.R.; D.A.R. provided the materials and analysis tools; D.A.R., A.B.Y.S. and A.A.C. performed formal analysis; D.A.R. writing —original draft preparation;

D.A.R. and A.A.C. writing—review and editing; funding acquisition, D.A.R. All authors have read and agreed to the published version of the manuscript.

**Funding:** Full funding was provided by the Ministry of Education and Culture, Republic of Indonesia (Competitive National Primary Research of Higher Education with contract number 1/E1/KP.PTNBH/2021).

**Institutional Review Board Statement:** Ethical review and approval were not required for this study. We use a non-invasive approach in collecting wild animal data.

**Informed Consent Statement:** Not applicable.

**Data Availability Statement:** Not applicable.

**Acknowledgments:** We thank Ahmad Abdul Aziz Fathur Rahman, Nafisatun Khasanah, and Trisna Risky Martiyani from Forest Resources Conservation and Ecotourism Study Program for their support in the field missions. We would also like to thank Jacqueline L. Sunderland-Groves from the University of British Columbia for constructive criticism and for improving the manuscript. We thank the staff of Gunung Halimun Salak National Park (GHSNP), Baluran National Park (BNP), and Natural Resources Conservation Agency (NRCA) of DKI Jakarta (Indonesian Ministry of Environment and Forestry) for authorising this research. In Cikaniki Resort Research Facility (CRRF), this research was authorised under permit number 9/P/TNGHS/2021 from GHSNP Authority. In Savana Bekol, this research was authorised under permit number S1/T.37/TU/KSA.6/5/2021 from the BNP Authority. In Pulau Rambut Wildlife Sanctuary, this research was authorised under permit number S.890/K.13/TU/Simaksi/07/2021 from the NRCA of DKI Jakarta Authority.

**Conflicts of Interest:** The authors declare no conflicts of interest.

## Appendix A

*Project Overview*

This study aimed to evaluate drone-based aerial surveys for censusing critical ecological aspects of wildlife communities in several ecosystem types that dominate tropical rainforest landscapes in Indonesia. Drones are reported to be potentially less physically laborious, less intrusive, and adequately accurate in wildlife monitoring than traditional ground-based surveys where surveyors walk across animals along specific transects. In this study, three conservation areas were selected based on the urgency of conservation and the unique ecosystem characteristics of each location, namely Pulau Rambut Wildlife Sanctuary (PRWS) with ecosystem type of coastal forest and mangrove forest, Baluran National Park (BNP) with ecosystem type of savanna, and Gunung Halimun Salak National Park (GHSNP) with montane forest ecosystem. Pulau Rambut Wildlife Sanctuary is a small island and breeding habitat for various types of birds. Pulau Rambut Wildlife Sanctuary is one of six areas designated as a Ramsar Site due to the large number of migratory bird species that live on this island. As many as 14 of the 48 bird species in PRWS are highly dependent on the ecosystem on this island. Furthermore, the site supports three internationally threatened bird species, especially the vulnerable Milky Stork (*Mycteria cinerea*), with one of the biggest breeding colonies of this species in Indonesia. Pulau Rambut Wildlife Sanctuary is located in the Thousand Islands (officially Kepulauan Seribu) in the Bay of Jakarta, where 13 rivers flow into this island. As a result, litter disposed of in Jakarta or the surrounding area is carried along Jakarta Bay to the Thousand Islands. This raises the problem of marine litter in these waters, especially in PRWS. During our study, in the northern part of the mangrove ecosystem on this island, various types of litter were found, such as wood, bamboo, plastic, and styrofoam. Marine litter accumulation was also found in the southeastern mangrove ecosystem. The accumulated litter can cause disturbance to the habitat and the conservation of waterbirds in PRWS. Waterbird monitoring in PRWS has previously been carried out using traditional ground-based surveys, but in practice, there are many obstacles and challenges, such as substrates that are affected by tides and high vegetation density which makes it difficult for direct observations to be carried out on land.

Baluran National Park (BNP) is one of the unique ecosystems in the tropical maritime continent of Indonesia. This landscape is also known as 'Africa van Java' due to its savanna ecosystem as a major natural resource that harbour various biodiversities such as banteng (*Bos javanicus*), Javan deer (*Rusa timorensis*), water buffalo (*Bubalus bubalis*), red muntjac (*Muntiacus muntjak*), Javan leopard (*Panthera pardus melas*), dhole (*Cuon alpinus*), and Javan green peafowl (*Pavo muticus muticus*). However, BNP faced multifaceted problems from socio-economics and environmental aspects—i.e., social conflicts in land management; invasive species; and wildfires. Some good conservation practices should be made to prevent the population decline of endemic and endangered species in BNP under various pressures. Identifying endangered species populations within BNP is requisite to understand further conservation management for the specific species. Therefore, finding out the best practices in detecting species is required for effective wildlife management and conservation. The ground-based survey approaches sometimes require extensive effort, time-consuming, and relatively challenging to encounter the species. Regular and reliable monitoring of wildlife species estimates is essential for the conservation work of any species. Besides, the use of drones for species surveys has improved the detectability of the species. Estimates of population size can be quite challenging for mammal's species, particularly in relatively wide areas. In our study, we present the experimental methods to assess species diversity in the savanna ecosystem based on RGB imagery and thermal imagery and machine learning approaches using thermal drone imageries for detecting species.

Gunung Halimun Salak National Park (GHSNP), especially Cikaniki Resort Research Facility (CRRF), is one of the essential areas in Indonesia in primate conservation. There are three endemic diurnal primates in Java that inhabit this area, such as the Javan gibbon (*Hylobathes moloch*), Javan langur (*Trachypithecus auratus*), Javan leaf monkey (*Presbytis comata*), and one nocturnal endemic primate (Javan slow loris *Nycticebus javanicus*). These four primates live arboreal in dense tree canopies, so the ground-based survey is often challenging to conduct. The low rate of encounters through ground surveys with endemic primates at CRRF causes very limited ecological data for the four primate species. Through this study, we evaluate the use of drones to monitor endemic primates in CRRF. This study is the first in Indonesia's tropical forests and was conducted to assess the behavioral response to the presence of drones in their habitat. As a reliable platform for wildlife monitoring, it is crucial to assess the behavioral responses shown by the focus animals before studies on other ecological aspects are carried out in the future.

## Appendix B

*Appendix B.1. Drone Matrice 300 RTK Specification*

**Table A1.** Drone *Matrice 300 RTK* applied at Pulau Rambut Wildlife Sanctuary and Baluran National Park.

| Hardware | Part of Drone | Detailed Information |
|---|---|---|
| Aircraft | Weight with one gimbal on the bottom | Approximately 3.6 kg (without battery) |
| | | Approximately 6.3 kg (with 2 batteries TB60) |
| | Dimension | 430 × 420 × 430 mm (folded) |
| | | 810 × 670 × 430 mm (takeoff) |
| | Diagonal length | 895 mm |
| | Maximum load | 2.7 kg |
| | Maximum takeoff weight | 9 kg |
| | Operating frequency | 24,000–24,835 GHz |
| | | 5725–5850 GHz |
| | RTK positioning accuracy (when on and installed) | 1.5 cm + 1 ppm (vertical) |
| | | 1 cm + 1 ppm (horizontal) |

| | | |
|---|---|---|
| | Maximum speed up | S mode: 6 m/s |
| | | P mode: 5 m/s |
| | Maximum speed down (vertical) | S mode: 5 m/s |
| | | P mode: 4 m/s |
| | Maximum speed down (tilt) | S mode: 7 m/s |
| | Maximum speed | S mode: 23 m/s |
| | | P mode: 17 m/s |
| | Maximum wind obstacle | 15 m/s |
| | Maximum flight duration | 55 min |
| | Navigation system | GPS + GLONASS + BeiDou + Galileo |
| | Operating temperature | −20 °C to +50 °C |
| Thermal Camera | Sensor | Uncooled VOx Microbolometer |
| | Lens | DFOV: 40.6° |
| | | Focal length: 13.5 mm (equivalent 58 mm) |
| | | Aperture: f/1.0 |
| | | Focus: 5 m to ∞ |
| | Sensor resolution | 640 × 512 |
| | Pixel pitch | 12 μm |
| | Spectral band | 8–14 μm |
| | Image size | 640 × 480 (4:3) |
| | | 640 × 360 (16:9) |
| | Scene range | High gain: −40° to +150 °C |
| | | Low gain: −40° to +550 °C |
| | Image format | R-JPEG |
| | Video format | MP4 |
| Visual Camera (Zoom Camera) | Sensor | Zoom camera: 1/1.7″ CMOS Sensor, 20 MP |
| | | Wide camera: 1/2.3″ CMOS Sensor |
| | Lens | Zoom camera |
| | | DFOV: approximately 66.6°–4° |
| | | Focal length: 6.83–119.94 mm |
| | | Aperture: f/2.8–f/11 (normal); f/1.6–f/11 (mode malam) |
| | | Focus: 1 m to ∞ (wide); 8 m to ∞ (telephoto) |
| | ISO range | Video: 100–25,600 (auto) |
| | | Gambar: 100–25,600 (auto) |
| Visual Camera (Wide Camera) | Sensor | Wide camera: 1/2.3″ CMOS Sensor, 12 MP |
| | Lens | DFOV: approximately 82.9° |
| | | Focal length: 4.5 mm |
| | | Aperture: f/2.8 |
| | | Focus: 1 m to ∞ |
| | ISO range | Video: 100–25,600 (auto) |
| | | Image: 100–25,600 (auto) |

*Appendix B.2. Drone Mavic 2 Enterprise Dual Specification*

**Table A2.** Drone Mavic 2 Enterprise Dual applied at Pulau Rambut Wildlife Sanctuary and Baluran National Park.

| Hardware | Part of Drone | Detailed Information |
|---|---|---|
| Aircraft | Weight without accessories | 899 g |
| | Maximum weight with accessories | 1100 g |
| | Dimension | 214 × 91 × 84 mm (folded) |

| | | |
|---|---|---|
| | | 322 × 242 × 84 mm (takeoff) |
| | Diagonal length | 354 mm |
| | Maximum speed up | 4–5 m/s |
| | Maximum speed down | 3 m/s |
| | Maximum speed without wind disturbance | 72 km/hour |
| | Maximum flight duration | 31 min (constant speed 25 km/hour) |
| | Navigation system | GPS + GLONASS |
| | Operating frequency | 2400–2835 GHz |
| | | 5725–5850 GHz |
| | Internal storage | 24 GB |
| Thermal Camera | Sensor | Uncooled VOx Microbolometer |
| | Lens | HFOV: 57° |
| | | Aperture: f/1.1 |
| | Sensor resolution | 160 × 120 |
| | Pixel pitch | 12 μm |
| | Spectral band | 8–14 μm |
| | Image size | 640 × 480 (4:3) |
| | | 640 × 360 (16:9) |
| | Accuracy | High gain: Max ±5% (typical) |
| | | Low gain: Max ±10% (typical) |
| | Scene range | High gain: −10° to +140 °C |
| | | Low gain: −10° to +400 °C |
| | Image format | JPEG |
| | Video format | MP4, MOV MPEG-4 AVC/H.264) |
| Visual Camera | Sensor | 1/2.3″ CMOS; Effective pixels: 12 M |
| | | FOV: approximately 85° |
| | | 35 mm format equivalent: 24 mm |
| | Lens | Aperture: f/2.8 |
| | | Focus: 0.5 m to ∞ |
| | ISO range | Video: 100–12,800 (auto) |
| | | Image: 100–1600 (auto) |

**Appendix C**

*Appendix C.1. Drone Flight Plan*

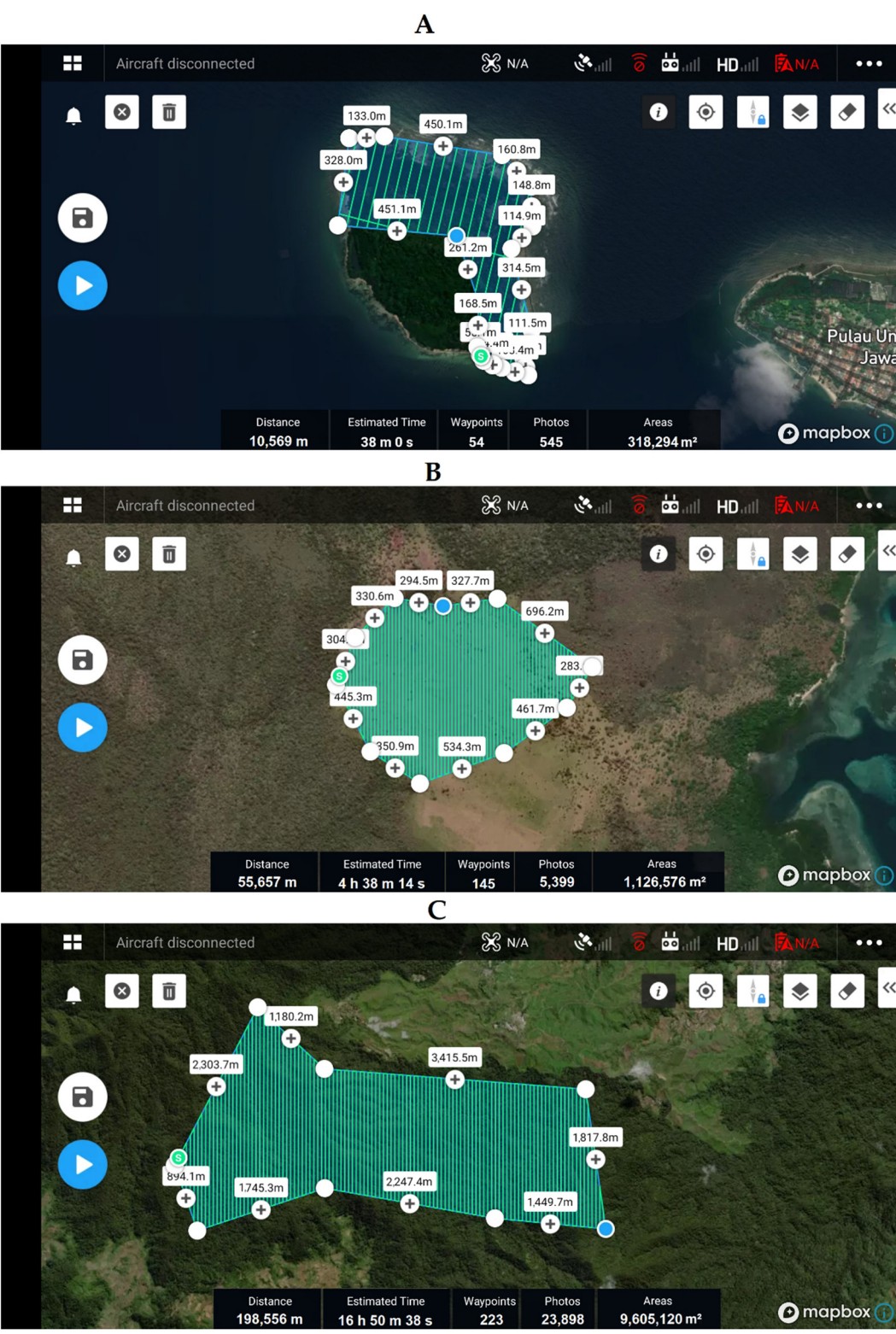

**Figure A1.** Example of flight missions using DJI Pilot version 1.9.0 in (**A**) coastal and mangrove forest in PRWS, (**B**) savanna in BNP, and (**C**) montane forest in GHSNP.

*Appendix C.2. Drone System and Operation Details*

**Table A3.** Flight mission data in Pulau Rambut Wildlife Sanctuary, Baluran National Park, and Gunung Halimun Salak National Park. See Figure A1 for visualization of flight plans.

| Location | Sensor | Time | Altitude (m) | GSD (cm/pixel) | Side Overlap | Frontal Overlap (%) |
|---|---|---|---|---|---|---|
| Pulau Rambut Wildlife Sanctuary | Zenmuse H20T Wide | 4 h 34 m 12 s | 120 | 4.14 | 70 | 80 |
| | | 4 h 47 m 18 s | 110 | 3.79 | 70 | 80 |
| | | 5 h 15 m 16 s | 100 | 3.45 | 70 | 80 |
| | | 5 h 57 m 54 s | 90 | 3.10 | 70 | 80 |
| | | 6 h 30 m 13 s | 80 | 2.76 | 70 | 80 |
| | | 7 h 24 m 33 s | 70 | 2.41 | 70 | 80 |
| | | 8 h 45 m 51 s | 60 | 2.07 | 70 | 80 |
| | | 10 h 13 m 37 s | 50 | 1.72 | 70 | 80 |
| | | 12 h 56 m 29 s | 40 | 1.38 | 70 | 80 |
| | | 16 h 58 m 9 s | 30 | 1.03 | 70 | 80 |
| | | 25 h 12 m 17 s | 20 | 0.69 | 70 | 80 |
| | | 33 h 34 m 22 s | 10 | 0.52 | 70 | 80 |
| Optimum detection | | 7 h 55 m 13 s | 65 | 2.24 | 70 | 80 |
| Baluran National Park | Zenmuse H20 T Wide | 1 h 22 m 49 s | 120 | 4.14 | 70 | 80 |
| | | 1 h 29 m 51 s | 110 | 3.79 | 70 | 80 |
| | | 1 h 38 m 34 s | 100 | 3.45 | 70 | 80 |
| | | 1 h 49 m 1 s | 90 | 3.10 | 70 | 80 |
| | | 2 h 1 m 45 s | 80 | 2.76 | 70 | 80 |
| | | 2 h 24 m 0 s | 70 | 2.41 | 70 | 80 |
| | | 3 h 13 m 24 s | 60 | 2.07 | 70 | 80 |
| | | 4 h 35 m 43 s | 50 | 1.72 | 70 | 80 |
| | | 7 h 8 m 29 s | 40 | 1.38 | 70 | 80 |
| | | 12 h 42 m 25 s | 30 | 1.03 | 70 | 80 |
| | | 28 h 59 m 1 s | 20 | 0.69 | 70 | 80 |
| | | 49 h 59 m 8 s | 10 | 0.52 | 70 | 80 |
| Optimum detection | | 4 h 35 m 43 s | 50 | 1.72 | 70 | 80 |
| Gunung Halimun Salak National Park | Mavic 2 Enterprise Dual | 19 h 55 m 45 s | 120 | 3.67 | 10 | 80 |
| | | 21 h 32 m 46 s | 110 | 3.36 | 10 | 80 |
| | | 23 h 32 m 30 s | 100 | 3.06 | 10 | 80 |
| | | 25 h 55 m 30 s | 90 | 2.75 | 10 | 80 |
| | | 28 h 56 m 12 s | 80 | 2.45 | 10 | 80 |
| | | 32 h 53 m 28 s | 70 | 2.14 | 10 | 80 |
| | | 38 h 6 m 23 s | 60 | 1.84 | 10 | 80 |
| | | 45 h 20 m 10 s | 50 | 1.53 | 10 | 80 |
| | | 56 h 15 m 15 s | 40 | 1.22 | 10 | 80 |
| | | 74 h 23 m 49 s | 30 | 0.92 | 10 | 80 |
| | | 110 h 38 m 39 s | 20 | 0.61 | 10 | 80 |
| | | 146 h 53 m 50 s | 10 | 0.46 | 10 | 80 |
| Optimum detection | | 49 h 16 m 47 s | 45 | 1.38 | 70 | 80 |

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
