# Peer review of "From Coastal to Montane Forest Ecosystems, Using Drones for Multi-Species Research in the Tropics"

_drones, doi:10.3390/drones6010006_

Round 1

Reviewer 1 Report

General comments

The study is interesting, rich and well done. Congratulations to the authors. My main concern is the structure, that I believe aligns more with a project report than a paper. The manuscript includes:

-Comparison of birds and nest detection, study of nest material and assessing breeding  using drones vs ground surveys and thermal vs RGB images

-Species Diversity, Species Detection and Model Evaluation by the Ground and Drone Surveys

-Species Detection and Model Evaluation (AI)

-Assessment of drone impact on primates behaviour

This is a compilation of 3 or 4 studies into one paper. I think that each of these questions can, and may be better addressed, independently. I would have personally tried to write more than one paper out of this work. Including more does not necessarily mean the result is better. Each of the studies has different data collection, analysis, and covers different subjects (behaviour, spatial distribution, wildlife inventory, conservation issues, AI assessment). I think that when putting all of it together it becomes a little bit confusing, as there is not a clear hypothesis but a collection of questions and results, and each individual study loses the value it has. I see a connection though: drones, wildlife, tropical environments.

The manuscript would benefit from an English proof reading because the writing style and some phrases are not correct.

Specific comments

L15-17- please indicate country, public may not know these places

Introduction:

I suggest reducing the 3 first paragraphs, that I find too long for an intro about the study area, biodiversity conservation and wildlife monitoring limitations, all quite general.

Introduction should be better focussed on what is known about drones for wildlife monitoring, thermal sensors and tropical environments, what is the current “gap” of knowledge and how this paper plans to address it, which is barely mentioned only at the end on L98-117.

Fig 1: it is not possible to read the text in the images, either correct this or delete it. Units should be International Metric System (m, km not miles).

L183 I suggest a section for describing the system “system description”

L187 “-30 min” ; “+5 km”why there is a + and minus sign here?

L198 “fiveteen” use fifteen

L238-240 “once” use “one”

Fig 2: I suggest making the small images larger so we can see the details of the “zoom” indicated with arrows. Also, include in the legend what is “B” (I see it s the green dots but for consistency, the legend should mention B).

L291 “indices” better use indexes

L334 “was used” use were used (plural)

Fig 14. What is exactly the value represented in y axis? “Behavioural response score” I may be lost on what the number means. Is this an average considering the “scores” described in table 1 and the number of individuals? How did you proceed if an individual showed more than one behaviour type?

L475 “unoccupied drone systems (UAVS)” please be consistent along the text and use only one term. Chose one: e.g. drones, UAS, UAVs, RPASs and keep it.

L771 I don t see the usefulness of appendix A for the paper

Author Response

Response to Reviewer 1 Comments

We would like to thank the reviewer for through reading of this manuscript and for the thougtful comments and constructive suggestions, which help to improve the quality of this manuscript. Our response follows.

General comments

Point 1: The study is interesting, rich and well done. Congratulations to the authors. My main concern is the structure, that I believe aligns more with a project report than a paper. The manuscript includes:

  • Comparison of birds and nest detection, study of nest material and assessing breeding using drones vs ground surveys and thermal vs RGB images
  • Species Diversity, Species Detection and Model Evaluation by the Ground and Drone Surveys
  • Species Detection and Model Evaluation (AI)
  • Assessment of drone impact on primate’s behaviour

This is a compilation of 3 or 4 studies into one paper. I think that each of these questions can, and may be better addressed, independently. I would have personally tried to write more than one paper out of this work. Including more does not necessarily mean the result is better. Each of the studies has different data collection, analysis, and covers different subjects (behaviour, spatial distribution, wildlife inventory, conservation issues, AI assessment). I think that when putting all of it together it becomes a little bit confusing, as there is not a clear hypothesis but a collection of questions and results, and each individual study loses the value it has. I see a connection though: drones, wildlife, tropical environments.

The manuscript would benefit from an English proof reading because the writing style and some phrases are not correct.

Response 1: I agree with you. Even though this paper is a compilation of 3 studies, in fact, some of the studies in our paper have similarities in terms of the survey techniques used and the hypothesis or fundamental question to be answered (how reliable are drones in wildlife monitoring and species conservation in the tropical areas when compared to ground-based surveys?). At the end of the manuscript, a general conclusion is presented that discusses the use of drones for multi-species research in each type of ecosystem and their relationship to one another. Furthermore, to create a link between the several studies in our paper, we have also improved the writing style of the entire manuscript, so it's more cohesive.

As our proposed title of this manuscript, this manuscript would like to provide several applications in using drones for biodiversity monitoring and conservation in a different ecosystem, particularly in the tropics We ensure that this study is one of a research article that provides comprehensive methods for species detection and monitoring for further potentially tackling global conservation issues. We thought that these various studies within the manuscript are also suitable for the journal's aim & scope and will bring a comprehensive example to the readers related to biodiversity monitoring methods and conservation. To assess the effectiveness of biodiversity monitoring, this study compares the utilization of drones with a conventionally terrestrial transect-based method that had been widely used for tropical rainforests ecosystems.

Finally, the manuscript has been carefully reviewed by an experienced editor whose first language is English and specializes in editing papers written by scientists whose native language is not English.

Specific comments

Point 2: L15-17- please indicate country, public may not know these places

Response 2: We have already added the country name in the location description in the abstract (third sentence, L19).

Point 3: Introduction:

I suggest reducing the 3 first paragraphs, that I find too long for an intro about the study area, biodiversity conservation and wildlife monitoring limitations, all quite general.

Introduction should be better focussed on what is known about drones for wildlife monitoring, thermal sensors and tropical environments, what is the current “gap” of knowledge and how this paper plans to address it, which is barely mentioned only at the end on L98-117.

Response 3: Thank you for your suggestion. We have modified the entire sentence in the first three paragraphs of the manuscript to make it more concise and structured (L34-88).

Point 4: Fig 1: it is not possible to read the text in the images, either correct this or delete it. Units should be International Metric System (m, km not miles).

Response 4: We have changed the metric map-scale into International Metric System, which is kilometres. We also have already corrected the figure text to make it more readable (L156-160).

Point 5: L183 I suggest a section for describing the system “system description”

Response 5: The system description of drones has already been described in Appendix B (L872-877).

Point 6:  L187 “-30 min”; “+5 km” why there is a + and minus sign here?

Response 6: We removed the "-" sign and changed the "+" sign with ">". The DJI Matrice 300 RTK with Thermal Camera DJI H20T and DJI Mavic 2 Enterprise Dual have a pilot-controlled range of a maximum of 7 km and 15 km, respectively (L201-202).

Point 7: L198 “fiveteen” use fifteen

Response 7: The correction has been made (L212).

Point 8: L238-240 “once” use “one”

Response 8: The correction has been made (L252-254).

Point 9: Fig 2: I suggest making the small images larger so we can see the details of the “zoom” indicated with arrows. Also, include in the legend what is “B” (I see it s the green dots but for consistency, the legend should mention B).

Response 9: The figure has been modified (L302-304).

Point 10: L291 “indices” better use indexes

Response 10: Thank you for your suggestion. We have considered changing the ‘indices’ word in L291. However, we thought that it would be more fit if we used the term ‘indices’ in the other sentences. Indices are used when referring to mathematical, scientific, or statistical contexts. Besides, indexes are usually used in reference to written documents (https://www.nasdaq.com/articles/indexes-or-indices-whats-the-deal-2016-05-12) (L308).

Point 11: L334 “was used” use were used (plural)

Response 11: The correction has been made (L351).

Point 12: Fig 14. What is exactly the value represented in y axis? “Behavioural response score” I may be lost on what the number means. Is this an average considering the “scores” described in table 1 and the number of individuals? How did you proceed if an individual showed more than one behaviour type?

Response 12: The figure has been modified, and the y-axis description has been clarified. Behavioral response scores are averaged by considering the "score" in Table 1 and the number of individuals of each primate species. If an individual showed more than one behaviour type, we used the highest number score for each animal in each survey method (ground vs drones) for statistical analysis during each observation period (L355-356).

Point 13: L475 “unoccupied drone systems (UAVS)” please be consistent along the text and use only one term. Chose one: e.g. drones, UAS, UAVs, RPASs and keep it.

Response 13: Done, we consistently use “drones” for the entire text.

Point 14: L771 I don t see the usefulness of appendix A for the paper

Response 14: Appendix A contains a description of the research project overview. Since this study discusses the use of drones for multi-species surveys in various ecosystem types, the authors feel the need to fully explain the urgency and why these three locations were chosen as the study locus to make it easier for readers to understand. And this is not found in any section, not in the Introduction, nor the Method. Appendix A describes the outline of the problem of why monitoring species using drones is carried out in the three locations and conservation issues in each location. To see the reliability, this method compared to conventional methods, namely ground-based surveys that have long been used to monitor wildlife in tropical areas. Still, this description is mainly associated with efforts to develop methods for monitoring and conserving biodiversity in each ecosystem type.

Reviewer 2 Report

This manuscript details the application of thermal and RGB photography with cameras carried by drones and the analysis undertaken by AI. I would recommend the manuscript for publication but I did find it heavy going to read and interpret and understand the key study findings. I have made specific comments on the manuscript but I have a few overall comments.

  1. I recommend the authors have the manuscript copy edited by a native English speaker to improve flow and clarity. This will greatly improve the discoverability and accessibility of what is technically a very good and robust study.

  1. The MS is a mish mash of various methodologies and multiple interpretations. I see the MS is it is really two separate studies crammed within a single MS. As such it sits uncomfortably between as a ecology paper and a methodological paper about drone technology, which is ultimately the focus of this journal As a reader I found myself confused in places if the text was referring to the nesting bird or the mammalian study, and that it jumped between describing ecology and validating new methods. A suggestion might be to break the study into two manuscripts one upon the Survey of birds and nests and the other upon the survey of mammals. These could be published side by side in the journal. However, I will leave that up to the editors to make that decision.

  1. IF authors decide to go with a single paper I would suggest removing some of the results. For example, the plastics in the birds nest. Sure that is of interest to bird ecologists but is it of interest to readers of drones. I recommend the readers figure out the key aims of the study. Then present these throughout each of the sections the MS>  

Author Response

Response to Reviewer 2 Comments

We would like to thank the reviewer for through reading of this manuscript and for the thoughtful comments and constructive suggestions, which help to improve the quality of this manuscript. Our response follows.

General comments

Point 1: This manuscript details the application of thermal and RGB photography with cameras carried by drones and the analysis undertaken by AI. I would recommend the manuscript for publication but I did find it heavy going to read and interpret and understand the key study findings. I have made specific comments on the manuscript but I have a few overall comments.

I recommend the authors have the manuscript copy edited by a native English speaker to improve flow and clarity. This will greatly improve the discoverability and accessibility of what is technically a very good and robust study.

The MS is a mish mash of various methodologies and multiple interpretations. I see the MS is it is really two separate studies crammed within a single MS. As such it sits uncomfortably between as an ecology paper and a methodological paper about drone technology, which is ultimately the focus of this journal as a reader I found myself confused in places if the text was referring to the nesting bird or the mammalian study, and that it jumped between describing ecology and validating new methods. A suggestion might be to break the study into two manuscripts one upon the Survey of birds and nests and the other upon the survey of mammals. These could be published side by side in the journal. However, I will leave that up to the editors to make that decision.

IF authors decide to go with a single paper I would suggest removing some of the results. For example, the plastics in the bird’s nest. Sure that is of interest to bird ecologists but is it of interest to readers of drones. I recommend the readers figure out the key aims of the study. Then present these throughout each of the sections the MS> 

Response 1: Thank you for your positive comments and suggestions. This manuscript would like to provide several applications in using drones for biodiversity monitoring and conservation in a different ecosystem, particularly in the tropics. We thought that these various studies within the manuscript are also suitable for the journal's aim & scope and will bring a comprehensive example to the readers related to biodiversity monitoring methods and their application for conservation. To create the connections between some of the studies in our paper as crucial as your comments, we have improved the writing style of the entire manuscript, making it more cohesive.

Regarding the substantial removal of the manuscript, we decided to bring out all of the analysis in three tropical ecosystems. We keep a special discussion of monitoring nesting materials in waterbirds to show the practical use of drones on conservation issues. This is a practical example of using drones for conservation purposes. While this is an additional result of the methods, we have developed for biodiversity monitoring using drones in tropical areas, this example of marine litter is just as crucial as establishing more effective and efficient waterbird monitoring methods. For instance, previous studies showed that marine litter influences bird populations and habitats [1,2]. This marine litter has become a global concern and issue, particularly for marine biodiversity and migratory waterbirds. Research conducted by [3] found a colony of black-legged Kittiwake (Rissa tridactyla) that uses marine litter in the form of plastic as a material to make nests. Another study on little black cormorant (Phalacrocorax sulcirostris) in Pulau Rambut Wildlife Sanctuary (PRWS) used synthetic rope as nesting material. These materials are in the form of synthetic rope [4]. In this study, the same thing was found.  The use of synthetic materials such as ropes is harmful to birds. Wildlife affected by marine litter is generally exposed to entanglement, eating marine debris, bioaccumulation, and changes in habitat function [5]. It was recorded that 536 species of animals were entangled in marine litter such as nets, fishing rods, bottles, and plastic materials [6]. Birds are one of the affected species. In 2015, it was recorded that 90% of birds in the world had eaten plastic waste [7]. In our study area, this can be seen in discovering the little black cormorant that died entangled in a rope in its nest [8]. These findings add to the long list of marine litter that may disturb the habitat and reduce the population of waterbirds in PRWS. Therefore, detecting plastic in the bird's nest is crucial for conservation strategies on the coastal bird species. However, the fundamental difference from the previous research is that our research is technically more practical. An observer does not need to climb a nest tree to check the nesting material, just fly a drone.

References:

  1. Ivonie, R.N.; Mardiastuti, A.; Rahman, D.A. Proposed classification of waste that landed on small island in Indonesia for the conservation of waterbird. IOP Conf. Ser. Earth Environ. Sci. 2020, 528, doi:10.1088/1755-1315/528/1/012012.
  2. Ivonie, R.N.; Mardiastuti, A.; Rahman, D.A. Daily accumulation rates of marine litter in Pulau Rambut Wildlife Sanctuary, Jakarta Bay, Indonesia. IOP Conf. Ser. Earth Environ. Sci. 2021, 771, doi:10.1088/1755-1315/771/1/012034.
  3. Hartwig, E.; Clemens, T.; Heckroth, M. Plastic debris as nesting material in a Kittiwake- (Rissa tridactyla)-colony at the Jammerbugt, Northwest Denmark. Marine Poll Bull. 2007, 54 (5), 595–597, doi: 10.1016/j.marpolbul.2007.01.027.
  4. Fithri, A. Analisa bahan sarang burung pecuk padi hitam (Phalacrocorax sulcirostris) di Suaka Magasatwa Pulau Rambut, Teluk Jakarta. Berita Biologi. 2007, 8(4), 241-247.
  5. Vegter, A.; Barletta, M.; Beck, C.; Borrero, J.; Burton, H.; Campbell, M.; Costa, M.; Eriksen, M.; Eriksson, C.; Estrades, A.; Gilardi, K.; Hardesty, B.; Ivar do Sul, J.; Lavers, J.; Lazar, B.; Lebreton, L.; Nichols, W.; Ribic, C.; Ryan, P.; Schuyler, Q.; Smith, S.; Takada, H.; Townsend, K.; Wabnitz, C.; Wilcox, C.; Young, L.; Hamann, M. Global research priorities to mitigate plastic pollution impacts on marine wildlife. Endang Species Res. 2014, 25, 225–247, doi:10.3354/esr00623.
  6. Kühn, S.; Bravo-Rebolledo, E.L.; van Franeker, J.A. Deleterious effects of litter on marine life. In: Bergmann, M.; Gutow, L.; Klages, M.; editor. Marine Anthropogenic Litter. New York (US): Springer, 2015, 75–116.
  7. Wilcox, C.; van Sebille, E.; Hardesty, B.D. Threat of plastic pollution to seabirds is global, pervasive, annepd increasing. Proc Natl Acad Sci USA. 2015, 112, 11899–11904. doi:10.1073/pnas.1502108112.
  8. Ivonie, R.N. Distribusi dan dampak sampah laut pada habitat burung merandai di Suaka Margasatwa Pulau Rambut. Master Thesis, IPB University, Bogor, Indonesia, 2020.

Moreover, the manuscript has been carefully reviewed by an experienced editor whose first language is English and specializes in editing papers written by scientists whose native language is not English.

Specific comments

Point 2: not sure what this means investigated prospective studies

Response 2: We have already changed the words to “prospective methods” (L17)

Point 3: were

Response 3: We have already changed “which” to “were” (L25).

Point 4: I Don’t know if the use of drones for survyeing animals and IA can be counted as novel anymore, there are literally hundreds of papers on this subject now. It would be better to here to identify what specific aspects of this were novel.

Response 4: We have already removed the “novel” word in the abstract (L28).

Point 5: this is probably debatable perhaps say arguably one of the most complex ecosystems.

Response 5: The correction has been made (L34).

Point 6: all ecosystems are vital for the animals that live there.

Response 6: Yes, I agree with you.

Point 7: what is fortress

Response 7: An area to preserve biodiversity. The sentence has been modified (L34-38).

Point 8: This needs to be re-edited by an english speaker.

Response 8: We have improved the writing style of the entire manuscript, including a check by an independent, native English speaker with a scientific background.

Point 9: this sentence makes no sense. I am not going to put these out again, but recommend this entire manuscript is edited. this will make it much more appealing to the broader readership.

Response 9: The correction has been made.

Point 10: this reads as if the ground surveys used the dual camera and not the UAV studies

Response 10: The correction has been made (L157-160).

Point 11: most of the mission flights were greater than the battery duration of the drone. Some of these flights were greater than 12 hours in duration, so it would be good for reader to understand how this was accounted for. I presume the drone came back and the battery was changed and it when back to the same location to commence the survey, but how was this done over time limits that ran over consequetive days, and how was light properties adjusted for.

Response 11: In drone flight, we prepare battery backup for up to 6 pairs. Every time the battery used during the flight runs out, we replace it with a spare battery and recharge the battery that has run out. In this way, we can use drones intensively.

Increasing the ISO and/or reducing the shutter speed when the light is low. Commonly, we fly the drone when the weather conditions are good (sunny or with little rainy). The weather was relatively good during our research because the research was carried out during the dry season.

Point 12: I would term as RGB and radiometric imagery throughout

Response 12: We have already changed the “visible imagery” term into “RGB imagery” throughout the manuscript.

Point 13: I am struggling to see the significance here, of the marine litter, when the paper is really about the methodology

Response 13: As our proposed title of this manuscript, this manuscript would like to provide several applications in using drones for biodiversity monitoring and conservation in a different ecosystem, particularly in the tropics. We ensure that this study is one of a research article that provides comprehensive methods for species detection and monitoring for further potentially tackling global conservation issues. We thought that these studies within the manuscript are also suitable for the journal's aim & scope and will bring a comprehensive example to the readers related to biodiversity monitoring methods and conservation. For complete reasoning regarding why we keep the topic of marine litter, please find it in the general comment section.  

Point 14: Is this only the % of total mammals?

Response 14: This percentage is for all species recorded by drones.

Point 15: The thermal cameras upon drones are uncooled and you will get thermal drift in tropical environments as the camera sensor hets up during the flight. This needs to be at least mentioned and how it was accounted for.

Response 15: We know that drones will produce heat during the flight. However, our thermal sensors (i.e., Uncooled VOx Microbolometer) are automatically calibrated and give a temperature bias of about ± 1 ºC. This bias temperature was acceptable for surveyed biodiversity, which was the warm-blooded species. We also have already addressed this issue in the manuscript.

Point 16: past tense change throughout

Response 16: The correction has been made to entire sentences (L634-647).

Reviewer 3 Report

Dear authors

The topic is timely, relevant, and underexplored.

Related work is discussed briefly. However, the paper seems to describe the process of work done.

It seems like story telling of conducted project than a research paper. The manuscript sounds technically.

The topic is interesting, the background research detailed, the description fully presented, but the paper generally lacks the evaluation of performed and the comparison with the other solutions. Because of it, it doesn't sound scientific.

Minor issues

Figures should have clear descriptions.

Figure 1. The descriptions on the drawings are completely illegible

Figure 4. The descriptions on the drawings are completely illegible (time, coordinates)

Figures 7, 8. Descriptions in the lower left corner are illegible Maybe they should be removed.

Figure 13. The same vertical scale (temperature) in both drawings (Javan deer and Water buffalo)  would give a clearer message to the reader

In my opinion (as non-native English speaker) the language and style are fine which make reception of the paper very pleasant.

In conclusion, the topic is timely, and the description of the work done is detailed. The article is more of a project description than a research paper.

Nevertheless, this work is unique because of the place and purpose of making measurements using new technologies such as unmanned aerial systems. Therefore, I can recommend the editor to consider publishing this manuscript in the Drones journal.

Author Response

Response to Reviewer 3 Comments

We would like to thank the reviewer for through reading of this manuscript and for the thougtful comments and constructive suggestions, which help to improve the quality of this manuscript. Our response follows.

General comments

Point 1: Dear authors

The topic is timely, relevant, and underexplored.

Related work is discussed briefly. However, the paper seems to describe the process of work done.

It seems like story telling of conducted project than a research paper. The manuscript sounds technically.

The topic is interesting, the background research detailed, the description fully presented, but the paper generally lacks the evaluation of performed and the comparison with the other solutions. Because of it, it doesn't sound scientific.

Response 1: We have corrected the writing style following the rules of scientific writing. Furthermore, in the conclusion section, we comprehensively compare and evaluation the use of drones with conventional ground-based survey methods and the potential use of drones for biodiversity monitoring. The several advantages of our approach are presented in the conclusions section and proved promising for wildlife surveys methods using drones in different ecosystems in tropical forests (L727-787).

Specific comments

Point 2: Figure 1: The descriptions on the drawings are completely illegible

Response 2: The figure has been modified. We have changed also the metric map scale into International Metric System, which is kilometres. We also have already corrected the figure text to make it more readable (L156).

Point 3: Figure 4: The descriptions on the drawings are completely illegible (time, coordinates)

Response 3: The image resolution in the manuscript is low, so it's not very clear. We will send the original image with the best resolution after the editor requests it. Because drones provide this precious data recording, we keep the information of the time, coordinates, and flight altitude.

Point 4: Figures 7, 8: Descriptions in the lower left corner are illegible Maybe they should be removed.

Response 4: The image resolution in the manuscript is low, so it's not very clear. We will send the original image with the best resolution after the editor requests it. Because drones provide this precious data recording, we keep the information of the time, coordinates, and flight altitude.

Point 5: Figure 13: The same vertical scale (temperature) in both drawings (Javan deer and Water buffalo) would give a clearer message to the reader

Response 5: The correction has been made to Figure 13. We have made a consistent y-scale for Javan deer and water buffalo (L486).

Point 6:  In my opinion (as non-native English speaker) the language and style are fine which make reception of the paper very pleasant.

Response 6: Thank you, to prepare a better manuscript, we send it to native English speakers.

Point 7: In conclusion, the topic is timely, and the description of the work done is detailed. The article is more of a project description than a research paper.

Nevertheless, this work is unique because of the place and purpose of making measurements using new technologies such as unmanned aerial systems. Therefore, I can recommend the editor to consider publishing this manuscript in the Drones journal.

Response 7: Thank you for your positive comments. We made various improvements, as other reviewers commented. We keep some parts that are technical and seem like project descriptions, this is to make it easier to understand the practical aspects of our research so that researchers, academics, or someone who works using drones to preserve biodiversity can easily understand and duplicate the technical issues.

Reviewer 4 Report

Please see the attached pdf file.

Author Response

Response to Reviewer 4 Comments

We would like to thank the reviewer for through reading of this manuscript and for the thougtful comments and constructive suggestions, which help to improve the quality of this manuscript. Our response follows.

General comment

The authors demonstrate the use of drones for effective surveys of various species and of their behavior, performing extensive campaigns in three study areas. The paper is longer than average research articles but is, in my opinion, interesting and useful for the ecology community; as such, the work gives insight into an interesting application of UAVs as survey tools, discussing advantages and limitations of the approach. The paper is well written and describes thoroughly the methodology adopted and the main results. I would recommend publication after some clarification about the following issues:

Specific comments

Point 1: Lines 129 and 153. The reference (Köppen 1936) should be included in the “References” section as Köppen, W. Das geographische System der Klimate. In: Handbuch der Klimatologie. I, Teil C, Gebrüder Borntraeger, Berlin, 1936.

Response 1: Thank you, the reference has been added (L964-965).

Point 2: Line 188. “km h-1” should read “?? ℎ−1” (superscript). Similarly, in Line 236 “m s-1” should read “? ?−1”.

Response 2: The correction has been made (L202 & L250).

Point 3: Line 198. “fiveteen” should read “fifteen”.

Response 3: The correction has been made (L212).

Point 4: Line 288. “Assesment” should read “Assessment”.

Response 4: The correction has been made (L305).

Point 5: Line 456. What are “W” and “p”? Is W the value of the Wilcoxon statistic? I understand that you have to determine if two groups of objects are different from one another in a statistically significant manner, but is Wilcoxon test the optimal choice?

Response 5: "W" is the Wilcoxon test statistic, while p-value or probability value is a number describing how likely it is that our data would have occurred by random chance. The Wilcoxon test is used to analyze the results of paired observations of two data, whether they are different or not. Despite its reliability when compared to other non-parametric statistical tests, we use the Wilcoxon test because it is a simple test that is commonly used to compare two continuous populations when only a small number of independent samples are available and the two original populations are not normal, this is consistent with the available data we have.

Point 6: Lines 457, 458. If “s=3.92 °C” and “s=2.99 °C” indicate the standard deviation, perhaps you could better use  or the abbreviation” s.d.”.

Response 6: The correction has been made. We have chosen s.d. to indicate the standard deviation (L482-483).

Point 7: Table 5. What is “Df”?

Response 7: Df stands for the degree of freedom. We have already changed ‘Df’ to ‘df’ to make it more standard to represent the degree of freedom in statistical analysis (L506).     

This manuscript is a resubmission of an earlier submission. The following is a list of the peer review reports and author responses from that submission.